# Decision Transformer under Random Frame Dropping

**Kaizhe Hu**[*]**, Ray Chen Zheng**[*]
Tsinghua University, Shanghai Qi Zhi Institute
`hkz22@mails.tsinghua.edu.cn`

**Yang Gao, Huazhe Xu**
Tsinghua Universtiy, Shanghai AI Lab, Shanghai Qi Zhi Institute

## Abstract

Controlling agents remotely with deep reinforcement learning (DRL) in the real world is yet to come. One crucial stepping stone is to devise RL algorithms that are robust in the face of dropped information from corrupted communication or malfunctioning sensors. Typical RL methods usually require considerable online interaction data that are costly and unsafe to collect in the real world. Furthermore, when applying to the frame dropping scenarios, they perform unsatisfactorily even with moderate drop rates. To address these issues, we propose Decision Transformer under Random Frame Dropping (DeFog), an offline RL algorithm that enables agents to act robustly in frame dropping scenarios without online interaction. DeFog first randomly masks out data in the offline datasets and explicitly adds the time span of frame dropping as inputs. After that, a finetuning stage on the same offline dataset with a higher mask rate would further boost the performance. Empirical results show that DeFog outperforms strong baselines under severe frame drop rates like 90%, while maintaining similar returns under non-frame-dropping conditions in the regular MuJoCo control benchmarks and the Atari environments. Our approach offers a robust and deployable solution for controlling agents in real-world environments with limited or unreliable data.

## 1 Introduction

Imagine you are piloting a drone on a mission to survey a remote forest. Suddenly, the images transmitted from the drone become heavily delayed or even disappear temporarily due to poor communication. An experienced pilot would use their skill to stabilize the drone based on the last received frame until communication is restored.

In this paper, we aim to empower deep reinforcement learning (RL) algorithms with such abilities to control remote agents. In many real world control tasks, the decision makers are separate from the action executor (Saha & Dasgupta, 2018), which introduce the risk of packet loss and delay during network communication. Furthermore, sensors such as cameras and IMUs are sometimes prone to temporary malfunctioning, or limited by hardware restrictions, thus causing the observation to be unavailable at certain timesteps (Dulac-Arnold et al., 2021). These examples lead to the core challenge of devising the desired algorithm: controlling the agents against frame dropping, i.e., a temporary loss of observations as well as other information.

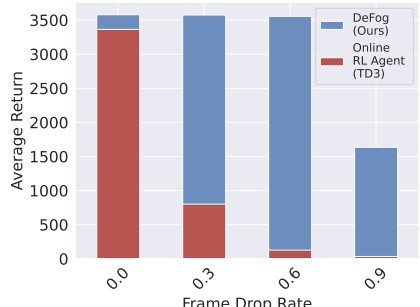

Figure 1: RL performance in the Hopper-v3 environment under different frame drop rates.

---
[*]equal contribution

Figure 1 illustrates how a regular RL algorithm performs under different frame drop rates. Our findings indicate that RL agents trained in environments without frame dropping struggle to adapt to scenarios with high frame drop rates, highlighting the severity of this issue and the need to find a solution for it.

This problem gradually attracts more attention recently: Nath et al. (2021) adapt vanilla DQN algorithm to a randomly delayed markov decision process; Bouteiller et al. (2020) propose a method that modifies the classic Soft Actor-Critic algorithm (Haarnoja et al., 2018) to handle observation and action delay scenarios. In contrast to the frame-delay setting in previous works, we try to solve a more challenging problem where the frames are permanently lost. Moreover, previous methods usually learn in an online frame dropping environment, which can be unsafe and costly.

In this paper, we introduce Decision Transformer under Random Frame Dropping (DeFog), an offline reinforcement learning algorithm that is robust to frame drops. The algorithm uses a Decision Transformer architecture (Chen et al., 2021) to learn from randomly masked offline datasets, and includes an additional input that represents the duration of frame dropping. In continuous control tasks, DeFog can be further improved by finetuning its parameters, with the backbone of the Decision Transformer held fixed.

We evaluate our method on continuous and discrete control tasks in MuJoCo and Atari game environments. In these environments, observations are dropped randomly before being sent to the agent. Empirical results show that DeFog significantly outperforms various baselines under frame dropping conditions, while maintaining performance that are comparable to the other offline RL methods in regular non-frame-dropping environments.

## 2 RELATED WORKS

### 2.1 CONTROL UNDER FRAME DROPPING AND DELAY

The loss or delay of observation and control is an essential problem in remote control tasks (Balemi & Brunner, 1992; Funda & Paul, 1991). In recent years, with the rise of cloud-edge computing systems, this problem has gained even more attention in various applications such as intelligent connected vehicles (Li et al., 2018) and UAV swarms (Bekkouche et al., 2018).

When reinforcement learning is applied to such remote control tasks, a robust RL algorithm is desired. Katsikopoulos & Engelbrecht (2003) first propose the formulation of the Random Delayed Markov Decision Process. Along with the new formulation, a method is proposed to augment the observation space with the past actions. However, previous methods (Walsh et al., 2009; Schuitema et al., 2010) usually stack the delayed observations together, which leads to an expanded observation space and requires a fixed delay duration as a hard threshold.

Hester & Stone (2013) propose predicting delayed states with a random forest model, while Bouteiller et al. (2020) tackle random observation and action delays in a model-free manner by relabelling the past actions with the current policy to mitigate the off-policy problem. Nath et al. (2021) build upon the Deep Q-Network (DQN) and propose a state augmentation approach to learn an agent that can handle frame drops. However, these methods typically assume a maximum delay span and are trained in online settings. Recently, Imai et al. (2021) train a vision-guided quadrupedal robot to navigate in the wild against random observation delay by leveraging delay randomization. Our work shares the same intuition of the train-time frame masking approach, but we utilize a Decision Transformer backbone with a novel frame drop interval embedding and a performance-improving finetuning technique.

### 2.2 TRANSFORMERS IN REINFORCEMENT LEARNING

Researchers recently formulate the decision making procedure in offline reinforcement learning as a sequence modeling problem using transformer models (Chen et al., 2021; Janner et al., 2021). In contrast to the policy gradient and temporal difference methods, these works advocate the paradigm of treating reinforcement learning as a supervised learning problem (Schmidhuber, 2019), directly predicting actions from the observation sequence and the task specification. The Decision Transformer model (Chen et al., 2021) takes the encoded reward-to-go, state, and action sequence as

input to predict the action for the next step, while the Trajectory Transformer (Janner et al., 2021) first discretizes each dimension of the input sequence, maps them to tokens, then predicts the following action's tokens with a beam search algorithm.

The concurrent occurrence of these works attracted much attention in the RL community for further improvement upon the transformers. Zheng et al. (2022) increases the model capacity and enables online finetuning of the Decision Transformer by changing the deterministic policy to a stochastic one and adding an entropy term to encourage exploration. Tang & Ha (2021) train transformer-based agents that are robust to permutation of the input order. Apart from these works, various attempts have been made to improve transformers in multi-agent RL, meta RL, multi-task RL, and many other fields (Meng et al., 2021; Xu et al., 2022; Lee et al., 2022; Reid et al., 2022).

## 3 METHOD

In this section, we first describe the problem setup and then introduce Decision Transformer under Random Frame Dropping (DeFog), a flexible and powerful method to tackle sporadic dropping of the observation and the reward signals.

### 3.1 PROBLEM STATEMENT

In the environment with random frame dropping, the original state transitions of the underlying Markov Decision Process are broken; hence, the observed states and rewards follow a new transition process. Inspired by the Random Delay Markov Decision Process proposed by Bouteiller et al. (2020), we define the new decision process as Random Dropping Markov Decision Process:

**Definition 1** (Random Dropping Markov Decision Process (RDMDP))
*An RDMDP could be described as $\mathcal{M} = \langle \mathcal{S}, \mathcal{A}, \mathcal{R}, \mu, \mathcal{P}, \mathcal{D}, \mathcal{O}_\mathcal{S}, \mathcal{O}_\mathcal{R} \rangle$, where $\mathcal{S}, \mathcal{A}$ are the state and action space, $\mathcal{P}(s_{t+1} \mid s_t, a_t)$ is the state transition possibility, $\mathcal{R}(s_t, a_t)$ is the reward function, $\mu(s_0)$ is the initial state distribution. $\mathcal{D}$ is the Bernoulli Distribution of frame dropping, $\mathcal{O}_\mathcal{S}$ is the function that emits the observation, and $\mathcal{O}_\mathcal{R}$ the function that emits the cumulative rewards. In the frame dropping setting, we assume the cumulative reward $R_t = \sum_{\tau=0}^{t} r_t$ is observed instead of the immediate reward $r_t$.*

*At each timestep $t$, a drop frame indicator $d_t \in \{0, 1\}$ is drawn from the distribution $\mathcal{D}$, with $d = 1$ indicating that the frame is dropped and $d = 0$ the opposite. The observed state $\hat{s}_t$ and the cumulative reward $\hat{R}_t$ of the timestep are updated by*

$$\hat{s}_t = \mathcal{O}_\mathcal{S}(s_t, \hat{s}_{t-1}, d) = d \cdot \hat{s}_{t-1} + (1 - d) \cdot s_t \tag{1}$$

$$\hat{R}_t = \mathcal{O}_\mathcal{R}(R_t, \hat{R}_{t-1}, d) = d \cdot \hat{R}_{t-1} + (1 - d) \cdot R_t \tag{2}$$

The observation and the cumulative reward repeats the last observed one $\hat{s}_{t-1}$ and $\hat{R}_{t-1}$ respectively if the current frame is dropped. If the current frame arrives normally, the observation and cumulative reward is updated. Note that the rewards are cumulated on the remote side, so the intermediate rewards obtained during the dropped frame are also added. Following the definition in the Decision Transformer where a target return $R_{\text{target}}$ is set for the environment, we define the real and observed reward-to-go as $g_t = R_{\text{target}} - R_t$, and $\hat{g}_t = R_{\text{target}} - \hat{R}_t$ respectively.

The goal of DeFog is to extract useful information from the offline dataset so that the agent can act smoothly and safely in a frame-dropping environment.

### 3.2 DECISION TRANSFORMER UNDER RANDOM FRAME DROPPING

We choose Decision Transformer as our backbone for its expressiveness and flexibility as an offline RL learner. To attack the random frame dropping problem, we adopt a three-pronged approach. First, we modify the offline dataset by randomly masking out observations and reward-to-gos during training, and dynamically adjust the ratio of the frames masked. Second, we provide a drop-span embedding that captures the duration of the dropped frames. Third, we further increase the robustness of the agent against higher frame dropping rates by finetuning the drop-span encoder and action predictor after the model is fully converged. A full illustration of our method is shown in Figure 2.

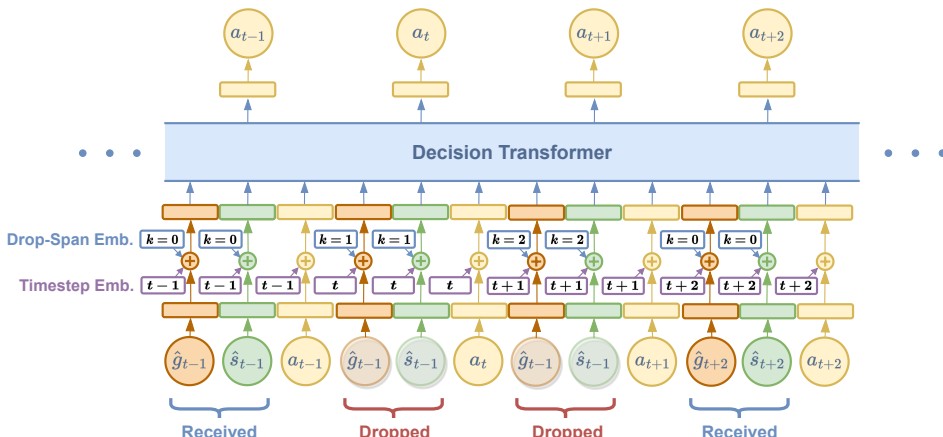

Figure 2: **Decision Transformer under Random Frame Dropping (DeFog).** Reward-to-go and state are repeated from the previous steps if the current frame is dropped. Timestep and drop-span embeddings, indicating the timestep and number of consecutive frame drops, are added onto the encoded reward-to-go and state before being sent to the Decision Transformer backbone. Since actions are not dropped, only the timestep embeddings are added to the encoded actions. The DT backbone outputs the predicted action embeddings, which is passed through a decoder to obtain the predicted actions.

### 3.2.1 DECISION TRANSFORMER BACKBONE

The Decision Transformer (DT) takes in past $K$ timesteps of reward-to-go $g_{t-K:t}$, observation $s_{t-K:t}$, and action $a_{t-K:t}$ as embedded tokens and predicts the next tokens in the same way as the Generative Pre-Training model (Radford et al., 2018). Let $\phi_g, \phi_s$ and $\phi_a$ denote the reward-to-go, state, and action encoders respectively, the input tokens are obtained by first mapping the inputs to a $d$-dimensional embedding space, then adding a timestep embedding $\omega(t)$ to the tokens.

Let $u_{g_t}, u_{s_t}$ and $u_{a_t}$ denote the input tokens corresponding to the reward-to-go, the observation, and the action of the timestep $t$ respectively, and $v_{g_t}, v_{s_t}$ and $v_{a_t}$ be their counterparts on the output token side. DT could be formalized as:

$$u_{g_t} = \phi_g(g_t) + \omega(t), \quad u_{s_t} = \phi_s(s_t) + \omega(t), \quad u_{a_t} = \phi_a(a_t) + \omega(t) \tag{3}$$

$$v_{g_{t-K}}, v_{s_{t-K}}, v_{a_{t-K}}, \ldots, v_{g_t}, v_{s_t}, v_{a_t} = \mathrm{DT}(u_{g_{t-K}}, u_{s_{t-K}}, u_{a_{t-K}}, \ldots, u_{g_t}, u_{s_t}, u_{a_t}) \tag{4}$$

Online decision transformer (ODT) by Zheng et al. (2022) enables online finetuning of the decision transformer models. We adopt the ODT model architecture because it has larger model capacity. Following their work, we also omit the timestep embedding $\omega(t)$ in the gym environments.

During training time, instead of directly predicting the action $a_t$, we follow the setting of the ODT to predict a Gaussian distribution for action from the output token of the state token inputs.

$$\pi_\theta(a_t \mid v_{s_t}) = \mathcal{N}(\mu_\theta(v_{s_t}), \Sigma_\theta(v_{s_t})) \tag{5}$$

The covariance matrix $\Sigma_\theta$ is assumed to be diagonal, and the training target is to minimize the negative log-likelihood for the model to produce the real action in the dataset $\mathcal{T}$:

$$J(\theta) = \frac{1}{K} \mathbb{E}_{(\mathbf{a},\mathbf{s},\mathbf{g}) \sim \mathcal{T}} \left[ -\sum_{k=1}^{K} \log \pi_\theta(a_k \mid v_{s_k}) \right] \tag{6}$$

### 3.2.2 TRAIN-TIME FRAME DROPPING

To prepare the model for frame dropping, we manually mask out observation and reward-to-go from the dataset. During the training stage, we specify an empirical dropping distribution $\hat{\mathcal{D}}$ and periodically sample "drop-masks" from it. A drop-mask is a binary vector of the same size as the

dataset and serves as the drop distribution $\mathcal{D}$ of an RDMDP. If a frame in the dataset is marked as "dropped" by the current drop-mask, the observation and reward-to-go of that frame are overwritten by the most recent non-dropped frame. We refer to the time span between the current frame and the last non-dropped frame as the drop-span of that frame.

One key consideration for the training scheme is the distribution $\hat{\mathcal{D}}$ of the drop-mask. A natural solution is to assume each frame has the same possibility $p_d$ to be dropped, and the occurrence of dropped frame is independent. Under this assumption, the stochastic process of whether each frame is dropped becomes a Bernoulli process. Additionally, we guarantee that the first frame for each trajectory is not dropped. After a certain number of training steps, the drop-mask is re-sampled from $\hat{\mathcal{D}}$ so that those dropped frames of the dataset could be used. We can also change $\hat{\mathcal{D}}$ as the training proceeds, for example, to linearly increase the $p_d$ throughout training. However, we empirically find that usually, a constant $p_d$ is sufficient.

### 3.2.3   DROP-SPAN EMBEDDING

In a frame dropping scenario, the agent must deal with the missing observation and reward-to-go tokens. Instead of dropping the corresponding tokens in the input sequence, we repeat the last observation or reward-to-go token and explicitly add a drop-span embedding to those tokens apart from the original timestep embedding $\omega(t)$.

Let $k_t$ denote the drop-span since the last observation of timestep $t$, the drop-span encoder $\psi$ maps integer $k_t$ to a $d$-dimensional token the same shape as the other observed input tokens. The model input with the drop-span embedding becomes:

$$u_{\hat{s}_t} = \phi_s(\hat{s}_t) + \psi(k_t) + \omega(t), \quad u_{\hat{g}_t} = \phi_g(\hat{g}_t) + \psi(k_t) + \omega(t), \quad u_{a_t} = \phi_a(a_t) + \omega(t) \quad (7)$$

Since the actions are decided and executed by the agent itself, they do not face the problem of frame dropping. The drop-span embedding is analogous to the timestep embedding in the Decision Transformer, but it bears the semantic meaning of how many frames are lost. Compared to the other indirect methods, the explicit use of drop-span embedding achieves better results. Detailed comparison could be found in Section 4.5.

### 3.2.4   FREEZE-TRUNK FINETUNING

The combination of the train-time frame dropping and the drop-span embedding is effective in making our model robust to dropped frames. However, we observed that in continuous control tasks, a finetuning procedure can further improve performance in more challenging scenarios.

Inspired by recent progress on prompt-tuning in natural language processing (Liu et al., 2021) and computer vision (Jia et al., 2022), we propose a finetuning procedure called "freeze-trunk finetuning" that freezes most of the model parameters during finetuning. The procedure involves finetuning the drop-span encoder $\psi$ and the action predictor $\pi_\theta$ after the model has converged.

During this stage, the training procedure is the same as that of the entire model. We draw drop-masks from $\hat{\mathcal{D}}$ to give the drop-span embeddings enough supervision, with the drop rate $p_d$ typically higher than in the main stage. While the number of training steps during this stage can be one-fifth of the main stage, the empirical results show that this procedure can improve the model's performance in higher dropping rates across multiple environments.

The whole training pipeline of our method could be found in Appendix A.

## 4   EXPERIMENTAL RESULTS

In this section, we describe our experiment setup and analyze the results. We compare our method with state-of-the-art delay-aware reinforcement learning methods, as well as online and offline reinforcement learning methods in multiple frame dropping settings. We first evaluate whether DeFog is able to maintain its performance as the drop rate $p_d$ increases. We then explore which key factors and design choices helped DeFog to achieve its performance. Finally, we provide insights of why DeFog can accomplish control tasks under severe frame dropping conditions.

## 4.1 EXPERIMENT SETUP

To comprehensively evaluate the performance of DeFog and baselines, we conduct experiments on three continuous control environments with proprioceptive state inputs in the gym MuJoCo environment (Todorov et al., 2012), as well as three discrete control environments with high-dimensional image inputs in the Atari games.

In each of the three MuJoCo environments, we use D4RL (Fu et al., 2020) which contains offline datasets of three different levels: expert, medium, and medium-replay. While in the three Atari environments, we follow the Decision Transformer to train on an average sampled dataset from a DQN agent's replay buffer (Agarwal et al., 2020). We train on 3 seeds and average their results in test time. We leave the detailed description of the settings to Appendix B.2.

During evaluation, we test the agents in an environment that has frame drop rates ranging from 0% to 90%. Results are shown by plotting the average return under 10 trials against test-time drop rate for different agents. The performance curve of our method is compared against various baselines:

- **Reinforcement learning with random delays (RLRD; Bouteiller et al., 2020).** RLRD is a method proposed to train delay-robust agents by adding randomly sampled delay as well as an action buffer to its observation space. RLRD has a maximum delay constraint and is not suited for discrete action tasks like Atari. In our frame dropping setting, we modify RLRD by limiting the delay value in the augmented observation to its maximum even if frames are still dropped. We compare our method to RLRD in the gym MuJoCo environment.

- **Twin-delayed DDPG (TD3; Fujimoto et al., 2018).** We train an online expert RL agent under regular non-frame-dropping settings using TD3 for continuous control tasks. We note that TD3 also has the privilege to interact with the environment.

- **Decision transformer (DT; Chen et al., 2021).** We train the vanilla DT using exactly the same offline datasets without the proposed components in Section 3.2.

- **Batch-constrained deep Q-learning (BCQ; Fujimoto et al., 2019).** BCQ is an offline RL method that aims to reduce the extrapolation error in offline RL by encouraging the policy to visit states and actions similar to the dataset.

- **TD3 + Behavioral cloning (TD3+BC; Fujimoto & Gu, 2021).** TD3+BC is built on top of TD3 to work offline by adding a behavior cloning term to the maximizing objective. Despite the simplicity, it is able to match the state-of-the-art performance.

- **Conservative Q-learning (CQL; Kumar et al., 2020).** CQL is a state-of-the-art model-free method which tries to address the issue of over-estimation in offline RL by learning a conservative Q function that lower-bounds the real one.

We leverage the implementation of Takuma Seno (2021) to train an offline agent in BCQ, TD3+BC, and CQL. We note that the online methods such as RLRD and TD3 are trained directly in the environment. Hence, their performance are invariable to different dataset types, and their curves are plotted repeatedly for comparison. We also include a DeFog without finetuning version to evaluate the effectiveness of freeze-trunk finetuning. Since TD3 cannot handle frame-dropping scenarios, we only plot it in the first row of the figures for better illustration. For a fair comparison, we assume the delay for RLRD is created by re-sending the dropped observations, which again has a probability $p_d$ of being lost. For baselines of the discrete control tasks, we train the offline RL agents until they reach the performance of DeFog under non-frame-dropping conditions, as we aim to evaluate how these methods preserve their performances under frame dropping settings.

## 4.2 EVALUATION IN THE CONTINUOUS CONTROL TASKS

We first evaluate our performance on three MuJoCo continuous control environments, namely Hopper-v3, HalfCheetah-v3, and Walker2d-v3. The results on each dataset are given in Figure 3.

We find that DeFog is able to maintain performance under severe drop rates. For example, the performance of the finetuned version on the Walker2d-Expert dataset barely decreases when the drop rate is as high as 80%. Meanwhile, the performance of the vanilla DT and TD3 agents come close to zero once the drop rate exceeds 67%.

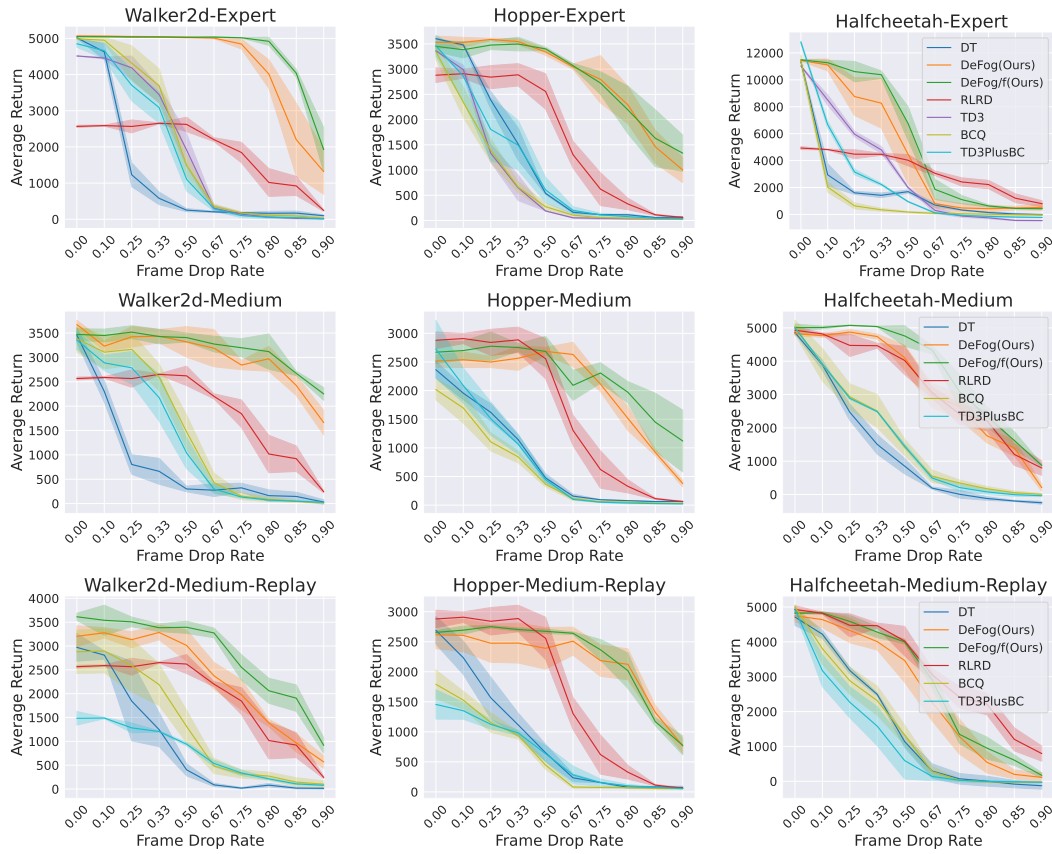

Figure 3: Performance on continuous control tasks. There are five baselines: a) TD3: the online TD3 agent, only included in the Expert datasets for a better scaling. b) DT: the offline Decision Transformer agent trained on the same dataset as DeFog. c): RLRD: the online RLRD agent that is optimized to deal with random frame delay. Since it's an online method there's no performance distinction between three kinds of datasets. d) BCQ, TD3PlusBC: other offline methods trained on the same dataset as DeFog. The full-fledged version of our method is indicated with DeFog/f (Ours), while the result of a non-finetuned version is indicated with DeFog (Ours) for comparison.

By looking at the starting point of the performance curves, we note that DeFog can achieve the same performance as the vanilla DT agent in non-frame-dropping scenarios. Despite a high train-time drop rate of 80% applied to DeFog, none of them are negatively affected when tested with a drop rate of 0%. As a comparison, the online RLRD method failed to achieve the same non-frame-dropping performance as other online baselines.

In the HalfCheetah-Expert setting, our method significantly outperforms the RLRD baseline with drop rates lower than 50%; however, in more extreme cases the RLRD takes over. RLRD, with its advantage of accessibility to the environment, was able to keep the performance better possibly due to HalfCheetah-Expert dataset's narrow distribution. In the medium and medium-replay datasets, DeFog is limited to data with less expertise, thus obtaining a reduced average return, but the overall performance of DeFog is comparable to that of RLRD.

We can also find that freeze trunk finetuning effectively improves DeFog's performance in various settings. In all 9 settings, the finetuned agent obtains better or at least the same results with the non-finetuned ones. The finetuning is especially helpful in high drop rate scenarios as shown in the Hopper-Medium and Walker2d-Expert settings. We highlight that the finetuning is done over the offline dataset without online interaction as well.

## 4.3 EVALUATION IN THE DISCRETE CONTROL TASKS

In this section, we evaluate our performance on three discrete control environments of Atari (Bellemare et al., 2013): Qbert, Breakout, and Seaquest. Following the practice of the Decision Transformer, we use 1% of a fully trained DQN agent's replay buffer for training. The results are shown in Figure 4. We find that DeFog outperforms the DT, BCQ and CQL baselines. We also find that in some environments, the performance of DeFog outperforms the Decision Transformer even under non-frame-dropping conditions. We believe that in these environments, using masked out datasets not only helps the agent to be more robust to frame dropping, but also makes the task more challenging in the sense that the agent needs to understand the environment dynamics better to give action predictions, which helps the agent make better decisions even when the frame drop rate is zero.

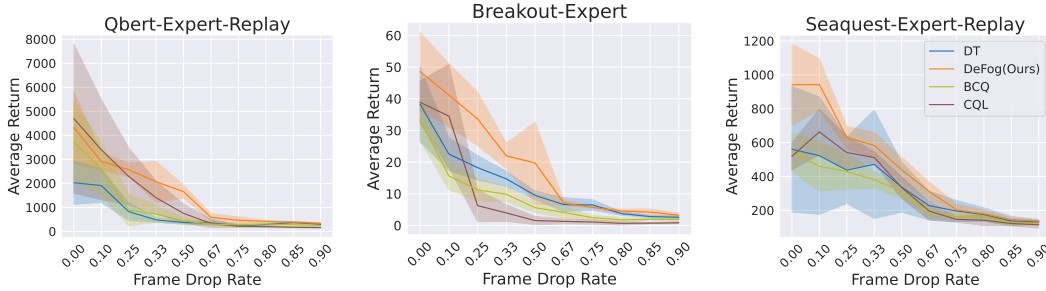

Figure 4: The performance in the discrete Atari game environments. DT is the offline Decision Transformer agent trained on the same dataset as DeFog. There are two other offline baselines: BCQ and CQL. Our method is indicated with DeFog (Ours).

## 4.4 VISUALIZED RESULTS

To gain a better understanding of our method, we visualize the results in the MuJoCo environment.

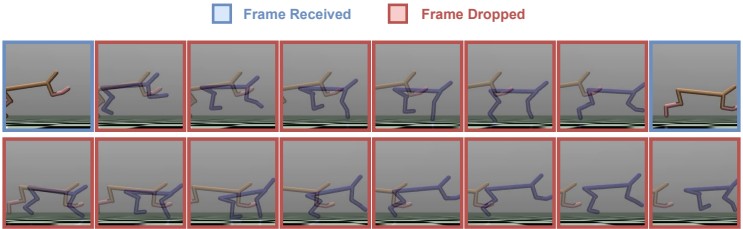

Figure 5: Visualization results of DeFog HalfCheetah agent under 90% frame drop rate. In each frame, the orange cheetah is the observed state $\hat{s}_t$, while the purple cheetah is the actual state $s_t$.

We first visualize the performance of an DeFog agent under a frame drop rate of 90% in Figure 5. The HalfCheetah agent (blue) is able to act correctly even if the observation (semi-transparent yellow) is stuck at 8 steps ago. Once a new observation comes in, the agent immediately adapts to the newest state and continues to perform a series of correct actions.

Building on top of the previous setting, we aim to exploit the capability of DeFog under extreme conditions by increasing the frame drop rate to 100%. In this way, the agent is only able to look at the initial observation and needs to make the rest of the decisions blindly. As shown in Figure 6, the HalfCheetah agent continues to run smoothly for more than 24 frames, demonstrating the Transformer architecture's ability to infer from the contextual history. We conjecture that such phenomenon is analogous to how humans perform a skill such as swinging a tennis racket without thinking about the observations.

## 4.5 ABLATION STUDY

We conduct ablation study on the drop-span embedding and freeze-trunk finetuning parts of DeFog.

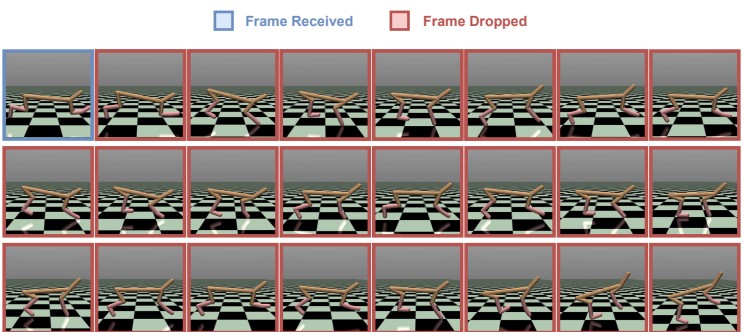

Figure 6: Visualization results of DeFog HalfCheetah agent under 100% frame drop rate. Only the very first observation is received. This scenario explores how far DeFog can go without any observation.

For the drop-span embeddings, we implement an alternative method that implicitly embeds the drop span information. Concretely, at each time step $t$, if the current frame is dropped, we change the corresponding timestep embedding $\omega(t)$ to the received one $\omega(t - k_t)$. Hence, for the implicit embedding method, we have $u_{s_t} = \phi_s(s_t) + \omega(t - k_t)$, $u_{g_t} = \phi_g(g_t) + \omega(t - k_t)$, $u_{a_t} = \phi_a(a_t) + \omega(t)$. In this way, the agent can infer the drop-span from the timestep embedding. As shown in Figure 7, we see that the proposed explicit drop-span embedding outperforms the implicit embedding, showing the effectiveness and necessity of explicitly providing the drop-span information to DeFog.

Now we ablate the freeze-trunk finetuning method by comparing the the same model with and without the finetuning stage. As shown in Figure 7, the finetuning outperforms the original model on all the continuous tasks. We believe that the performance gain in complex continuous control tasks is due to the crucial modules (i.e., action predictor and drop-span encoder) further adjusting themselves after the Decision Transformer backbone has converged. We provide more ablation studies in Appendix C.

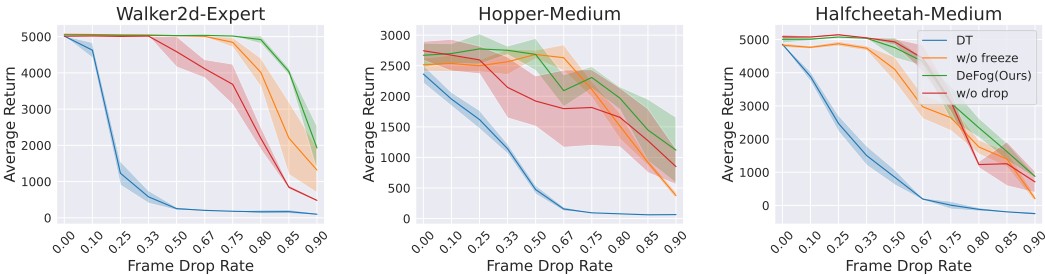

Figure 7: Ablation study of the drop-span embedding and the finetuning stage. "w/o freeze" means without freeze-trunk finetuning, and "w/o drop" stands for without drop-span embedding. We find that the proposed methods are effective in the continuous control tasks.

## 5    CONCLUSION

In this paper, we introduce DeFog, an algorithm based on Decision Transformer that addresses a critical challenge in real-world remote control tasks: frame dropping. DeFog simulates frame dropping by randomly masking out observations in offline datasets and embeds frame dropping timespan information explicitly into the model. Furthermore, we propose a freeze-trunk finetuning stage to improve robustness to high frame drop rates in continuous tasks. Empirical results demonstrate that DeFog outperforms strong baselines on both continuous and discrete control benchmarks under severe frame dropping settings, with frame drop rates as high as 90%. We also identify a promising future direction for research to handle corrupted observations, such as blurred images or inaccurate velocities, and to deploy the approach on a real robot.

ACKNOWLEDGMENT

This work is supported by the Ministry of Science and Technology of the People´s Republic of China, the 2030 Innovation Megaprojects "Program on New Generation Artificial Intelligence" (Grant No. 2021AAA0150000). This work is also supported by a grant from the Guoqiang Institute, Tsinghua University.

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

# A    ALGORITHM DETAILS

The overall algorithm of DeFog could be summarized as in Algorithm 1; for the hyperparameters we use, please refer to Appendix B.2.

---

**Algorithm 1** Decision Transformer under Random Frame Dropping (DeFog)

---

**Require:** update_interval {interval to update the drop-mask}
**Require:** n_updates, n_transitions {number of updates, number of transitions in the dataset}
**Require:** **sample**($\mathcal{S}$, $n$) {sample n elements uniformly from $\mathcal{S}$}
**Require:** **cumcount**(flags) {count cumulative number of 0's before current position (inclusive)}
**Require:** **train**(rtg, obs, act, timestep, dropspan, freeze_trunk) {train the DT model}
**Require:** batch_size, act_buffer, obs_buffer, rtg_buffer, context_length
  **for** freeze_trunk $\in$ {**true**, **false**}, **do**
    $N \leftarrow$ n_updates
    **while** $N \neq 0$ **do**
      $M \leftarrow$ update_interval
      **if** freeze_trunk **then**
        $M \leftarrow \lfloor M/5 \rfloor$
      **end if**
      drop_mask $\leftarrow$ **sample**({**true**, **false**}, n_transitions) {whether each frame is dropped}
      dropspans $\leftarrow$ **cumcount**(drop_mask) {duration of frames dropped}
      **while** $M \neq 0$ **do**
        selected_index $\leftarrow$ **sample**({$0, 1, ..., $n_transitions}, batch_size)
        timestep $\leftarrow$ selected_index
        dropspan $\leftarrow$ dropspans[selected_index]
        dropped_index $\leftarrow$ selected_index $-$ dropspan
        rtgs $\leftarrow$ rtg_buffer[dropped_index: dropped_index $+$ context_length]
        observations $\leftarrow$ obs_buffer[dropped_index: dropped_index $+$ context_length]
        actions $\leftarrow$ act_buffer[selected_index: selected_index $+$ context_length]
        **train**(rtgs, observations, actions, timestep, dropspan, freeze_trunk)
        $M \leftarrow M - 1$
      **end while**
      $N \leftarrow N - 1$
    **end while**
  **end for**

---

# B    EXPERIMENT DETAILS

## B.1    DATASETS AND SETUP

### B.1.1    GYM MUJOCO

We use the D4RL dataset (Fu et al., 2020) that contains data collected by SAC agents. There are three datasets for each environment – expert, medium, and medium-replay. The expert dataset is collected by a fully-trained expert policy, while the medium dataset is collected by an agent about half the performance of the expert. Medium-replay includes the trajectories in a medium agent's buffer, and is the most diverse dataset with the lowest average return. Details of the different datasets are provided in Table 1.

### B.1.2    ATARI

For Atari game environments, we use the DQN Replay Dataset (Agarwal et al., 2020), which is collected from the replay buffer of a DQN agent during training of these Atari games. Following the practice of the Decision Transformer, we only use a small portion of the dataset: 1% of the whole dataset, which is 500 thousand of the 50 million transitions observed by an online DQN agent. We define three kinds of datasets for each game as well – expert, medium, and expert-replay. The expert

| Dataset | No. Trajectories | No. Timesteps | Average Returns | Best Returns |
|---|---|---|---|---|
| Halfcheetah-Expert | 1000 | 1000 000 | 10656.43 | 11252.04 |
| Halfcheetah-Medium | 1000 | 1000 000 | 4770.33 | 5309.38 |
| Halfcheetah-Medium Replay | 202 | 202 000 | 3093.29 | 4985.14 |
| Hopper-Expert | 1027 | 999 494 | 3511.36 | 3759.08 |
| Hopper-Medium | 2186 | 999 906 | 1422.06 | 3222.36 |
| Hopper-Medium Replay | 2041 | 402 000 | 467.3 | 3192.93 |
| Walker2d-Expert | 1000 | 999 214 | 4920.51 | 5011.69 |
| Walker2d-Medium | 1190 | 999 995 | 2852.09 | 4226.94 |
| Walker2d-Medium Replay | 1093 | 302 000 | 682.7 | 4132 |

Table 1: D4RL Gym MuJoCo Datasets Sizes and Returns

and the medium dataset are collected from the DQN agent's buffer during the final and medium training stages, while the expert-replay dataset is sampled evenly from the whole replay buffer.

## B.2 HYPERPARAMETER SETTINGS

### B.2.1 GYM MUJOCO

For the gym MuJoCo environment, we use the same model architecture as the Online Decision Transformer (Zheng et al., 2022). While the Online Decision Transformer uses different training parameters for each environment, we keep most of the training parameters the same among different environments.

| Hyperparameter | Value |
|---|---|
| Number of layers | 4 |
| Number of attention heads | 4 |
| Embedding dimension | 512 |
| Training context length $K$ | 20 |
| Dropout probability | 0.1 |
| Activation function | ReLU |
| Gradient norm clip | 0.25 |

(a) Architecture Parameters

| Hyperparameter | Value |
|---|---|
| Learning rate | 1e-4 |
| Weight decay | 1e-3 |
| Batch size | 256 |
| Total training steps | 1e5 |
| Finetune training steps | 2e4 |
| Learning rate warmup steps | 1e4 |
| Drop-mask update interval | 100 |

(b) Training Parameters

Table 2: Common Parameters for Gym MuJoCo

For each dataset, we specify a target reward, and report the combination of train time drop-rate and finetuning drop-rate. The environment and dataset related paramenters are as follows:

| Environment | Dataset | Target Reward | Training Drop Rate | Finetuning Drop Rate |
|---|---|---|---|---|
| HalfCheetah | Expert | 12000 | 0.5 | 0.5 |
| HalfCheetah | Medium | 12000 | 0.8 | 0.8 |
| HalfCheetah | Medium Replay | 12000 | 0.8 | 0.8 |
| Hopper | Expert | 4000 | 0.9 | 0.9 |
| Hopper | Medium | 4000 | 0.5 | 0.8 |
| Hopper | Medium Replay | 4000 | 0.8 | 0.8 |
| Walker2d | Expert | 5000 | 0.9 | 0.9 |
| Walker2d | Medium | 5000 | 0.8 | 0.8 |
| Walker2d | Medium Replay | 5000 | 0.8 | 0.8 |

Table 3: Dataset Specific Parameters for Gym MuJoCo

### B.2.2 ATARI

For the Atari environments, we use the same model architecture as the Decision Transformer since the Online Decision Transformer doesn't perform experiments on these environments. The hyperparameters are as follows:

| Hyperparameter | Value |
|---|---|
| Number of layers | 6 |
| Number of attention heads | 8 |
| Embedding dimension | 64 |
| Training context length $K$ | 30 |
| Dropout probability | 0.1 |
| Activation function | ReLU |
| Gradient norm clip | 1.0 |

(a) Architechture Parameters

| Hyperparameter | Value |
|---|---|
| Learning rate | 6e-4 |
| Weight decay | 0.1 |
| Batch size | 128 |
| Total training steps | 1e5 |
| Finetune training steps | 2e4 |
| Learning rate warm up steps | 1e4 |
| Drop-mask update interval | 1000 |

(b) Training Parameters

Table 4: Common Parameters for Atari

The environment and dataset related parameters are given in Table 5. Linear increasing frame drop rate means that the drop rate is linearly increased from the start to end values.

| Environment | Dataset | Target Reward | Training Drop Rate | Finetuning Drop Rate |
|---|---|---|---|---|
| Qbert | Expert Replay | 14000 | 0.4 | 0.5 |
| Seaquest | Expert Replay | 1150 | 0–0.8 Linear Increase | 0.8 |
| Breakout | Expert Replay | 90 | 0–0.8 Linear Increase | 0.8 |

Table 5: Dataset Specific Parameters for Atari

## C  SUPPLEMENTARY RESULTS

In this section, we present further experimental results on the different components and settings of the DeFog model. Since we want to show the change in the agent's performance as the frame drop rate increases, the results are presented by the performance curves of the average return against the frame drop rate. To make the results more descriptive but not overwhelming, the three most representative curves are selected for most of the settings, while the descriptions and analyses of the results are based on all the settings.

### C.1  DECISION TRANSFORMER BACKBONE

**Training a non-Decision Transformer Model on Masked-out Datasets**  DeFog simulates the frame dropping scenario by using a masked dataset with frames intentionally hidden from the agent. This is tightly integrated with our drop-span embedding in the DeFog model, as the drop-span information must be supervised and conveyed into the hidden representations. To determine the Decision Transformer architecture's contribution to DeFog's strong results in frame dropping scenarios, we conduct an experiment with the TD3+BC (Fujimoto & Gu, 2021) baseline to train on a masked dataset. We use a masking rate of 50%, which is on par with or lower than that we use in DeFog.

The results are shown in Figure 8. The TD3+BC trained with a masked dataset is able to perform slightly better than the normal TD3+BC agent under higher frame drop rates in the Walker2d and Hopper environments. However, it collapses in the HalfCheetah environment. Although the average return improves slightly using a masked dataset, it is still nowhere close to the performance of DeFog. We believe this shows that the use of a masked dataset alone is not enough for DeFog's achievement.

**Reconstruction of Frames During Training**  The Decision Transformer architecture can issue three different types of tokens, corresponding to the next action, state, and reward-to-go respectively. While the authors of Decision Transformer only let the model predict the actions, it may be helpful to infer the actual state when the observation is dropped. With this motivation, we conduct experiments to see if letting the model predict the actual state or reward-to-go has a positive impact on the model's performance. We evaluate the influence of predicting state, the reward-to-go, and both on all nine settings of the MuJoCo tasks and report the results for three of them in Figure 9.

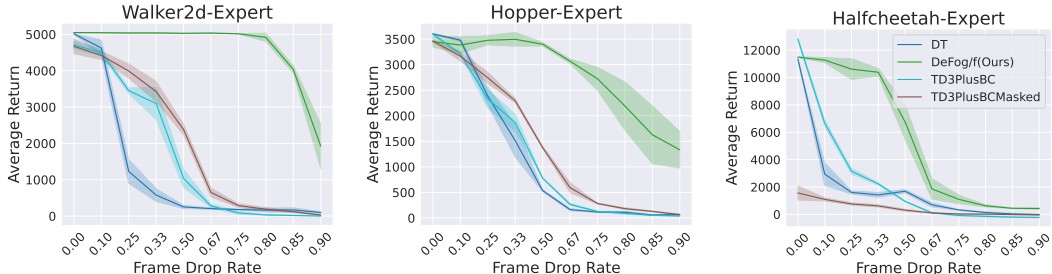

Figure 8: Ablation study on training a non-Decision Transformer based method with a masked-out dataset. "TD3PlusBC" is TD3+BC trained on a perfect uncorrupted dataset, while "TD3PlusBCMasked" denotes training a TD3+BC agent with a masked-out dataset.

We find it somewhat surprising that the performance of the model deteriorates significantly in four of the nine environments (HalfCheetah-Expert, Walker2d-Medium-Replay, Walker2d-Expert, and HalfCheetah-Medium-Replay) when only state prediction is applied. Only predicting the reward-to-go apart from the actions doesn't hinder the performance as much; predicting both of them doesn't affect the performance in general. We suspect this is due to the lack of supervision on the reward signal, which is exacerbated when the model is forced to predict both the state and action signals. In the original setting, where only the actions were predicted, and in the last setting, where all three tokens were predicted, this type of imbalance isn't as pronounced.

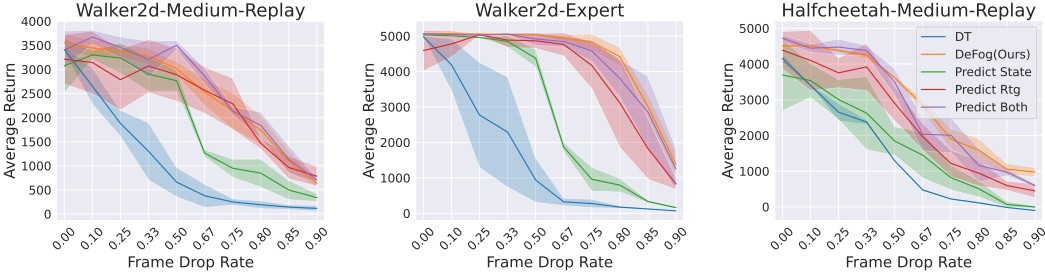

Figure 9: Ablation study on dropping the action together with the observation and reward-to-go.

## C.2 TRAIN-TIME FRAME DROPPING

In this section, we examine more carefully on the train-time frame dropping scheme, specifically the interval for resampling the drop-mask, the placeholder for dropped frames, the random process for generating the drop-mask, and the content to drop from the observation.

**Frame Dropping Mask** DeFog periodically samples and updates a drop-mask that decides which frames in the dataset are marked as dropped. By doing so, DeFog can take advantage of the full dataset and avoid overfitting the current un-masked dataset. To further explore the learning ability of DeFog, we conduct the experiment where the drop-mask never updates. In this way, the dropped frames which take up 50% to 90% of the dataset are never seen by DeFog during training .

Results in Figure 10 show that the performance of DeFog without update is similar to the original version using the full dataset and still outperforms other baselines, implying that DeFog is able to learn even when the dropped frames are never seen. We note that the performance degrades on the medium-replay datasets of Halfcheetah and Hopper environments. One potential reason is the relatively small volume of these datasets. As shown in Table 1, while the expert and medium datasets contain around 1M timesteps of data, the medium replay datasets have only 200k–400k. In the case of Halfcheetah-Medium-Replay, the number of non-dropped steps is only 8% of the Halfcheetah-Medium dataset.

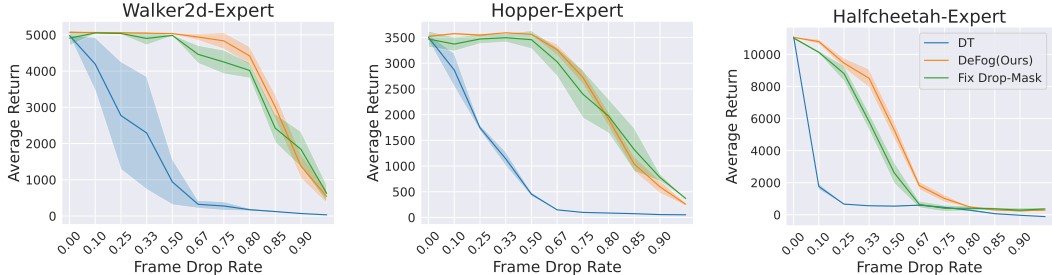

Figure 10: Ablation study on fixing the frame drop-mask. "Fix Drop Mask" indicates fixing the drop-mask throughout training.

**Placeholder for Dropped Frames**  During train-time frame dropping, if a frame is marked as dropped, DeFog follows a simple and intuitive approach to replace both the observation and the reward-to-go of that frame to the most recent non-dropped ones. We explore the following substitutions for the dropped frames:

- Adding noise to the dropped frames. This could be interpreted as stimulating the evolution of the unknown real states. For each step, we sample from a Gaussian noise distribution which is estimated from all the changes between consecutive observations in the dataset. When frames are dropped successively, the Gaussian noises add up to form a new Gaussian distribution. We use a scale factor of 0.1 and 0.5 to experiment the influence of the noise intensity.
- Simply replacing the dropped frames with zeros.
- Replacing the embedding of those dropped tokens to a specific learnable [MASK] token. We trial on two settings, one where the dropped observation and dropped reward-to-go share the same [MASK] token, and the other where the two tokens are separate.

The results are presented in Figure 11 and 12.

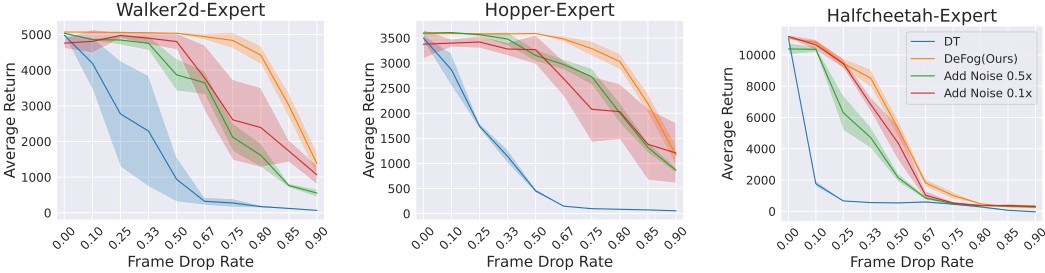

Figure 11: Ablation study on adding noise to those dropped frames. "Add Noise 0.1x" and "Add Noise 0.5x" denote a noise scaling factor of 0.1 and 0.5, respectively.

The results show that for adding noises, neither the scaling factor of 0.1 nor 0.5 helps with DeFog's performance. We find that increasing the noise intensity simply makes performance worse. In datasets such as Hopper-Medium and Walker-Expert, the deterioration is more noticeable.

In the case where we replace the dropped frames with learnable [MASK] tokens, both settings have performance better than vanilla Decision Transformer but worse than DeFog. We do not find this result surprising as a single learnable mask cannot carry enough information for all the dropped frames, while the previous frame that DeFog uses would be similar to the current dropped frame.

Finally, replacing dropped frames with zeros does not result in a much better performance than the vanilla Decision Transformer, as the zero token basically provides no information. Interestingly, the zero-masked version performs better than the learnable-token version. When using zero tokens for the dropped observation and reward-to-go, the transformer backbone receives nothing more than the drop-span embedding, which turns out to better convey the information needed for control than that when added with a learnable token.

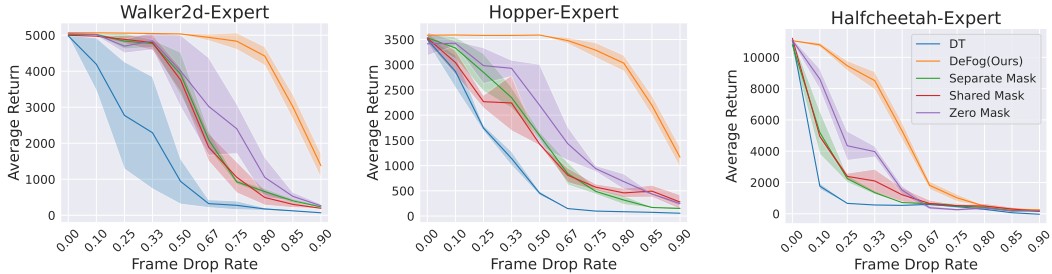

Figure 12: Ablation study on replacing the dropped frames with different kinds of [MASK] tokens. "Separate Mask" denotes that the observation and the reward-to-go do not share the same [MASK] token, while "Shared Mask" indicates the opposite. The "Zero Mask" simply consists of all zeros.

**Frame Dropping Process**   The binary sequence of whether each frame is dropped can be viewed as a random process. In DeFog, we use a fixed drop rate $p_d$ as the probability for any single frame to be dropped, which results in a Bernoulli process for dropping frames. To explore other kinds of dropping processes, we conduct experiments on the setting of frame dropping being a Markov process. The probability of the next frame being dropped is no longer a constant value $p_d$, but instead follows the transition matrix:

$$\mathbf{P} = \begin{bmatrix} 1 - p_1 & p_1 \\ 1 - p_2 & p_2 \end{bmatrix}$$

The matrix could be interpreted in the follow manner: given the current frame is not dropped, the probability for the next frame to be dropped is $p_1$; if the current frame is dropped, then the probability for the next frame being dropped is $p_2$. The reason for choosing a Markov process is it resembles the behavior in communication scenes where frames are dropped chunk by chunk rather than frame by frame. If $p_1 = p_2$, then the situation degenerates to a Bernoulli process.

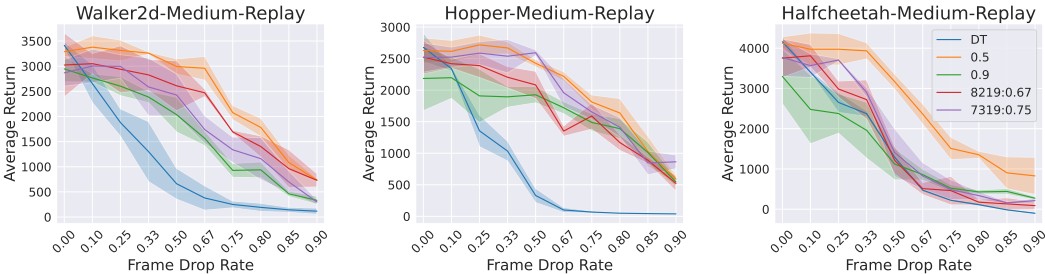

Figure 13: Ablation study on using a Markov Process for frame dropping. "0.5" and "0.9" represent using a Bernoulli Process of $p_d = 0.5$ and $p_d = 0.9$. "8219:0.67" denotes $p_1 = 0.2$, $p_2 = 0.9$, with a steady distribution of frame dropping probability $0.67$. Similarly, "7319:0.75" means $p_1 = 0.3$, $p_2 = 0.9$ with a steady distribution of frame dropping probability $0.75$.

Our experimental results, given in Figure 13, show that when comparing the Markov dropping process to the Bernoulli one, the agent trained under a Markov dropping process with drop probability $p_2$ performs similar to that with a Bernoulli dropping process under $p_d$, and this pattern is somewhat universal no matter what $p_1$ is. We find this result to abide with the fact that in a frame dropping setting, the moments where frames are dropped affect the overall performance more. If we fix $p_2$ and change $p_1$, we find that in general the less $p_1$ is, the better the performance. This is not surprising as decreasing $p_1$ would imply that there are more timesteps of consecutive undropped frames where the agent can leverage and make better decision.

**Dropping the Action**   In the training of DeFog, we drop the observation and the reward-to-go of the frames marked by the drop-mask, while remaining the action of those frames untouched. We

perform an extra experiment where the actions are masked out alongside the state and reward-to-go, and results show that the performance is negatively affected as shown in Figure 14.

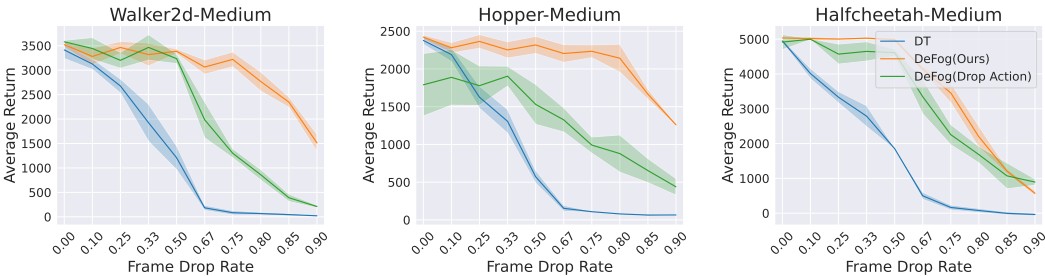

Figure 14: Ablation study on dropping the action together with the observation and reward-to-go.

### C.3 Drop-Span Embedding and Freeze Trunk Finetuning

**Explicit Drop-Span Encoder and Finetuning** Figure 15 contains the full ablation results of Figure 7, showing the performance of the ablated models on all datasets.

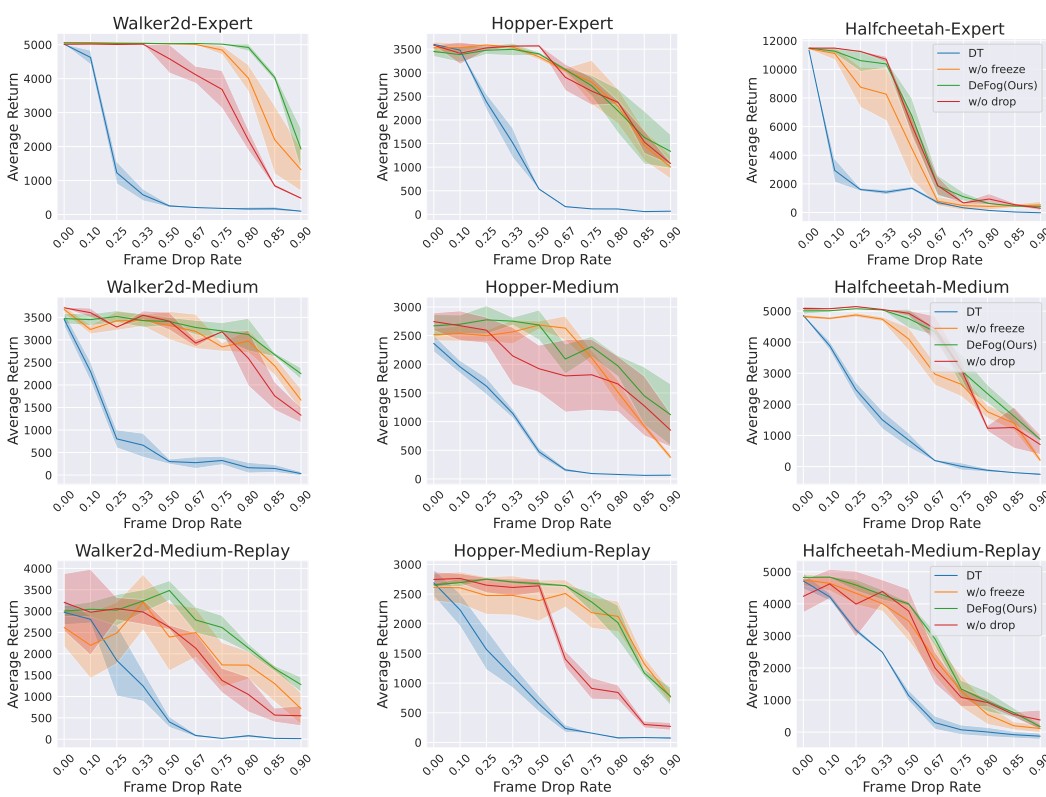

Figure 15: Ablation results on the explicit drop-span encoder and freeze-trunk finetuning in all 9 gym MuJoCo environments. The label "w/o freeze" stands for without freeze-trunk finetuning, while "w/o drop" denotes using the implicit embedding method.

The use of explicit drop-span embedding was able to improve performance over implicit embedding by a huge margin in 4 datasets. For the other 5 datasets, using implicit embedding all led to deterioration in performance as well, though not so significant. We believe this shows that the information leveraged by a DeFog agent is the relative rather than the absolute timestep of when the last frame was observed. The longer the drop-span of the current frame, the less it should be considered in action prediction, and the action history could be a better reference for decision-making. We conclude

that critical information like the drop-span needs to be explicitly given, and performance would be hindered even if the agent can work out the number by simple arithmetic.

**Removing Drop-Span and Timestep Embeddings**    Both the explicit drop-span encoder and the implicit embedding try to convey the drop-span information to the agent. We also conduct experiments on totally removing this piece of information, by using a normal timestep embedding without any other kind of drop-span embedding. The agent no longer receives information on how many frames are dropped. Finally, we perform the experiment of removing the timestep embedding but keeping the drop-span embedding. As mentioned above, the explicit drop-span encoder only gives information on the relative time span of dropped frames, not the actual timestep.

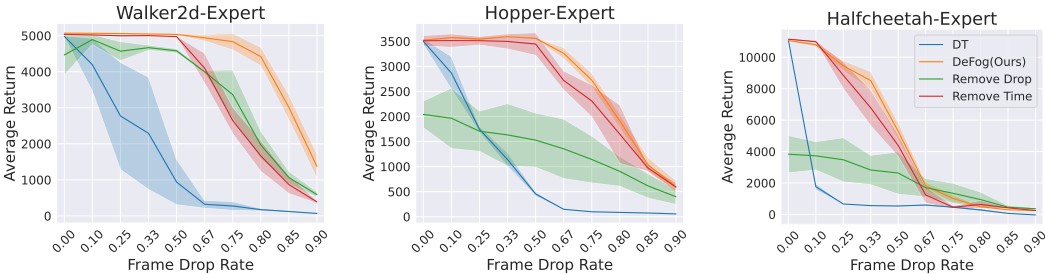

Figure 16: Ablation study on removing the drop-span information. "Remove Drop" denotes removing the drop-span embedding without using implicit method, while keeping the timestep embedding; "Remove Time" indicates removing the timestep embedding, while keeping the explicit drop-span embedding.

The results are given in Figure 16, and the performance is degraded upon both of the embeddings' removal. We find drop-span embedding to be the key factor in DeFog. Meanwhile, the removal of the timestep embedding does not cause a severe drop in performance. Under non-frame-dropping conditions, the Online Decision Transformer also conducted the experiments of removing timestep embeddings and found that performance was not heavily affected. As suggested by Zheng et al. (2022), this could be due to the timestep information deduced from the reward-to-go signal, making the lack of timestep embedding no longer fatal.

**Finetuning Individual Components**    DeFog currently finetunes the action predictor and the drop-span encoder. For a better understanding of the finetuning stage, as well as the functions of specific elements in the model, we conduct experiments on finetuning these components separately. The results are given in Figure 17.

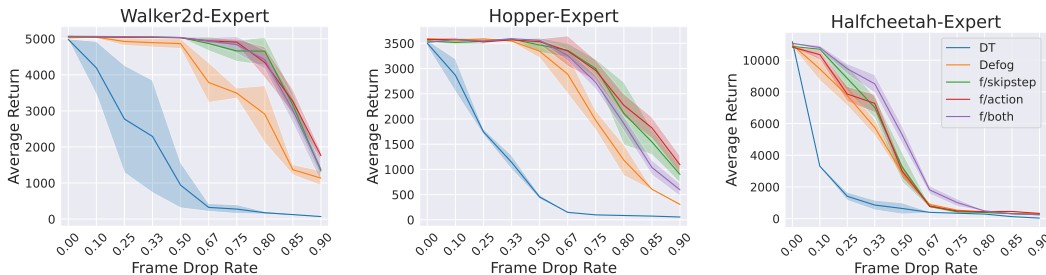

Figure 17: Ablation study on separately finetuning the components of DeFog. "DeFog" denotes not finetuning anything. "f/skipstep", "f/action", "f/both" stand for finetuning the drop-span encoder, the action predictor, and both, respectively.

We find that only finetuning the drop-span encoder gives slightly better performance on the Walker2d-Medium-Replay, Walker2d-Expert, and HalfCheetah-Medium-Replay datasets. While on the other datasets, for example the Hopper-Medium-Replay, finetuning both the drop-span encoder and the action predictor led to better results. In general, none of the finetuning methods significantly outperform their counterparts. We believe this is understandable as the action predictor and the drop-span encoder are both key components of the DeFog model.

