# OpenReview forum: "Decision Transformer under Random Frame Dropping"
_ICLR.cc/2023/Conference — ICLR 2023 poster_

### Official Review · Reviewer_dgXg · 2022-10-24

**Confidence:** 4
**Correctness:** 3
**Technical Novelty And Significance:** 2
**Empirical Novelty And Significance:** 3
**Recommendation:** 6

**Clarity, Quality, Novelty And Reproducibility:**

# Clarity
The paper is fairly well-written but the clarity could be improved in certain places. Some important details about the experimental setup are missing.

1. For example, it is not immediately clear whether the authors consider frame dropping during data collection, training, finetuning, at test time or a combination of these. I recommend adding more details about the precise setting in the introduction to avoid confusion. The experimental setup should be further clarified since it's not very clear. You seem to assume no frame dropping during data collection, but a changing distribution of frame dropping during training. Is this true? Do you keep the same percentage of frame dropping during training? Does this percentage change at test time or during the finetuning stage? Does evaluation happen online for all methods? What changes at test time, is it only the frame dropping percentage and is that fixed?

2. The authors mention making deep RL (DRL) agents robust to frame dropping in the abstract and introduction, but this is misleading since they don't propose a DRL algorithm, but rather an algorithm that learns from offline datasets which is a very different settings. Hence, those claims are incorrect.

3. On page 2, you mention the drop-span embeddings but don't define them. I don't think this is a common term in the literature so I suggest defining it the first time you mention it to avoid confusion.

4. Other details such as the number of runs used to compute the results in the plots are missing. This is an important detail so should be mentioned in the main paper rather than appendix.

5. I found Figure 2 to be somewhat hard to read. I'm not sure why you chose to add those lines on top of the timestep and drop-span embeddings. If they don't serve a particular purpose, I suggest removing them to make it more readable.

6. I found the sentence on page 3 "the optimization goal of our method is to imitate the oracle policy that have access to the true state in a frame-dropping environment through learning from the trajectories generated by the oracle" quite confusing. It seems to be contradicting your motivation of learning under frame-rating which would imply that you don't have access to the oracle and perfect trajectories, but rather the data was collected under frame dropping scenarios. I don't know when this assumption is realistic since it's as if you're saying "we can't collect uncorrupted data but here we assume we have access to uncorrupted data".

# Quality
As mentioned above, the paper needs more work in order to warrant acceptance at the ICLR conference. In particular, 1) the problem setting needs to be clarified, 2) the experimental setup needs to be realistic and ensure it is addressing the problem we care about rather than making unrealistic assumptions and leading to trivial conclusions, 3) more appropriate baselines need to be added for a fair comparison, 4) more ablations needs to be added for a better understanding of where the gains are coming from.

# Novelty
The ideas in this paper are not particularly novel. The proposed approach is a combination of well-known ideas but this exact instantiation hasn't been proposed before as far as I know. The authors cite related work and places their contribution in the context of the broader literature. I would say novelty is not a major problem with this paper, so I wouldn't use this as a reason to reject it if the other (more important) issues are resolved.

# Reproducibility
I didn't see any mention of the code or plans to open source it. While the paper contains reasonably details about the methods and experimental setup, this is not enough to easily reproduce their results. Do the authors plan to open source their code and if so, what is the timeline for that? This is an important factor in my decision.


**Strength And Weaknesses:**

# Strengths
1. well-motivated problem of some practical interests
2. the paper is easy to read overall
3. evaluation on two widely-used domains

# Weaknesses
### 1. the setting doesn't make sense to me and in light of this, the results are not surprising at all
- In the intro, you motivate this work by saying that in the real-world, often times observations are dropped or corrupted. However, as far as I understand, you collect an "oracle" dataset where no observations are dropped (or corrupted). Then, you randomly corrupt them in order to learn a more robust algorithm. But the whole point of solving this realistic setting is that you cannot collect / don't have access to such perfect / uncorrupted trajectories, isn't it? To me it seems like you motivated your work with one setting but you proceed to solve a completely different setting with totally unrealistic assumptions for the problem you claim to solve.
- It is not surprising at all that if you mask out observations you will learn a more robust policy than DT which learns with full observations so when observations are masked out it has no way of knowing how to deal with that.
- I believe the setting you should be considering (as motivated by the intro) is one in which your offline dataset contains dropped frames but your goal is still to learn a robust policy from them, perhaps by randomly masking out other frames.
- In addition, you seem to be changing the dropped frames as mentioned at the top of page 5 "sample drop-mask periodically from it". Again, this indicates that your method has access to oracle trajectories collected without any frame dropping and then you randomize which ones are dropped, which of course will make the method more robust.

### 2. the comparisons seem unfair an not very relevant
- First of all, I don't think the comparison between online RL methods like TD3 / RLRD and DT / DeFog is fair. DT and DeFog have access to the oracle trajectories without any dropped frames (which DeFog perturbs to make the policy more robust). If I understand correctly, TD3 and RLRD collect data online under the frame-dropping regime so they can never see the same trajectory with and without certain frames dropped in order to learn what it should do in that situation. In contrast, DeFog can randomly drop frames so it gets to see the oracle. Thus, this is an unfair comparison. To make it more fair, you need to basically collect the offline dataset used to train DT an DeFog under the same frame dropping setting / environment as used for training your online RL baselines.
- In addition, DeFog has explicit access to the timestep and drop-span embedding. Do TD3 and RLRD have access to this information? There is no mention of this but if they don't, the comparison is unfair because DeFog again has privileged information.
- In general, comparing with online RL is quite strange and hard to make a fair comparison because it collects its own data so the results will be highly dependant on the offline dataset used by DT / DeFog (even assuming that your offline data is collected under the same conditions as the online data).
- Given the above, the paper is incomplete without more fair comparisons with strong offline RL baselines trained on the same datasets as DT / DeFog such as IQL, CQL, offline DQN, and even BC (behavioral cloning) or BC transformer. These experiments could shed more light into whether the benefits are coming from the transformer architecture, offline learning, masking approach, dataset or something else. It could be the case that this masking also help offline RL methods which would make your approach even more general.

### 3. needs more baselines and ablations to better understand where the gains are coming from
- I would also like to see ablations of DeFog that doesn't use the drop-span embedding at all, one that doesn't use the timestep embedding at all, and what that doesn't use any of them. This would help in better assessing the contribution of each part of the architecture. Otherwise, again it is difficult to know where the gains are coming from.
- For the finetuning part, I would like to see ablations that only finetune the action predictor and only finetune the drop-span encoder.
- It would also be interesting to see what happens if instead of repeated prior frames, you replace them with zeros or an average of the N previous frames, a random frame from the previous N, or the previous frame with some added noise. This would shed more light into that particular choice of the algorithm. Does repeating the frame help because it adds some noise or is using the prior frame more helpful than adding other types of noise?

### 4. the setting and experimental setup isn't very clear, some important details are missing
- please see the section on Clarity below


# Minor
1. Why doesn't finetuning help in Atari?
2. Why doesn't your approach help as much in Atari?
3. Why is the variance of DT so large on Atari?
4. In Figure 7, it looks like removing the explicit drop embedding only hurts performance on Walker. Do you understand why that is? Does it mean it's not so important after all? In the text, you claim this is an important contribution, but the results seem mixed. I recommend running this ablation on more environments for a more robust conclusion (and try to understand why / when it helps).
5. How many seeds do you use?

**Summary Of The Paper:**

This paper addresses the problem of learning with random frame dropping. They propose to augment the decision transformer by randomly masking out observations in the offline dataset and explicitly adding the timespan of frame dropping as input to the transformer. They demonstrate that their approach, called DeFog, outperforms two online RL baselines under significant frame-dropping in Atari and MuJoCo.

**Summary Of The Review:**

While this paper addresses a problem of practical interest and is quite well-written, the experimental setup has some significant flaws including unfair comparisons, so it is difficult to draw any useful insights in the current stage. I recommend the authors to take into account the received feedback and resubmit an improved version of this work to a future conference.

---

> ### Author Response · Authors · 2022-11-17
> **Response to Reviewer dgXg(4/4)**
>
> ### Minor Issues
>
> 1. *Why doesn't finetuning help in Atari?*
> 2. *Why doesn't your approach help as much in Atari?*
> 3. *Why is the variance of DT so large on Atari?*
>
> **A**: We believe that the differences between DeFog's performance on MuJoCo and Atari come from the distinctions between the two environments, including the high dimensional observation space, different transition dynamics, and several other reasons. As the other reviewers have asked similar questions, we provide a detailed analysis and explanation in the "Discussion on the Atari Environments" section of the general response.
>
> Despite these problems, DeFog is still able to achieve better performance than a number of offline baselines in frame dropping scenarios. In the updated version of our paper, two more common offline RL methods, CQL and BCQ are proven to be weaker than DeFog as well. It’s worth noting that our method also provide a framework for solving frame dropping problem, and could be applied onto other models beyond the Decision Transformer to overcome these challenges.
>
> 4. *In Figure 7, it looks like removing the explicit drop embedding only hurts performance on Walker. Do you understand why that is? Does it mean it's not so important after all? In the text, you claim this is an important contribution, but the results seem mixed. I recommend running this ablation on more environments for a more robust conclusion (and try to understand why / when it helps).*
>
> **A**: Figure 8 in Appendix C.3 contains our full ablation results on the drop-span embedding, which were run on all 9 continuous control datasets. The label “w/o drop” means that we use implicit embedding instead, which we have described in paragraph 2 of Section 4.5. Drop-span embedding was able to improve performance over implicit embedding by a huge margin in 4 of these datasets: Walker2d-expert, Hopper-medium, Walker2d-medium-replay, and Hopper-medium-replay. For the other 5 datasets, implicit embedding all led to harm in performance as well, though not so significant. In addition, in 5 of the 9 datasets (Walker2d-expert, Walker2d-medium, Walker2d-medium-replay,  Hopper-medium, Hopper-medium-replay), performance was especially improved on high frame drop rates such as 0.8, 0.85, 0.9. The result shows that drop-span information is important and should be explicitly encoded into the model, as we have analyzed in the paper.
>
> 5. *How many seeds do you use?*
>
> **A**: We use 3 seeds for training and 10 trials for evaluation. We have added this clarification to Section 4.1.
>
> ---
>
> ### References
>
> [1] S. Fujimoto and S. S. Gu, “A Minimalist Approach to Offline Reinforcement Learning.” arXiv, Dec. 03, 2021. Accessed: Nov. 14, 2022. [Online]. Available: http://arxiv.org/abs/2106.06860
>
> [2] A. Kumar, A. Zhou, G. Tucker, and S. Levine, “Conservative Q-Learning for Offline Reinforcement Learning.” arXiv, Aug. 19, 2020. [Online]. Available: http://arxiv.org/abs/2006.04779
>
> [3] S. Fujimoto, D. Meger, and D. Precup, “Off-policy deep reinforcement learning without exploration,” in Proceedings of the 36th international conference on machine learning, Jun. 2019, vol. 97, pp. 2052–2062. [Online]. Available: https://proceedings.mlr.press/v97/fujimoto19a.html
>
> [4] L. Chen et al., “Decision Transformer: Reinforcement Learning via Sequence Modeling.” arXiv, Jun. 24, 2021. [Online]. Available: http://arxiv.org/abs/2106.01345
>
> [5] M. Janner, Q. Li, and S. Levine, “Offline Reinforcement Learning as One Big Sequence Modeling Problem.” arXiv, Nov. 28, 2021. [Online]. Available: http://arxiv.org/abs/2106.02039

---

> > ### Comment · Reviewer_dgXg · 2022-11-24
> > **Post-Rebuttal Response**
> >
> > Thank you for the detailed response.
> >
> > The authors have clarified most of the misunderstandings and I now believe the comparisons are fair and the setting makes sense. The authors have also added new experiments with the suggested baselines and ablations, making the empirical evaluating more thorough. I really appreciate that the authors took into account the received feedback, responded to each concern, and updated the paper accordingly.
> >
> > I think it would still be useful to add variants of the other baselines with and without frame dropping during training for a more complete empirical evaluation. Given the multiple misunderstandings regarding the setting and experimental setup, the writing could further be improved.
> >
> > In conclusion, I am now leaning towards acceptance and will increase my score to 6.

---

> > > ### Author Response · Authors · 2022-11-28
> > > **Further Discussion**
> > >
> > > Dear Reviewer dgXg, thank you for acknowledging our modifications and clarifications to the paper. We will keep polishing the paper writing in the final version. For your suggestion on more experiments with and without frame dropping, we conduct additional experiments on all the baselines in the Gym environments. The results show that masking frames alone is not enough to overcome the challenge of frame dropping. Both the train-time frame dropping and the drop-span embedding are necessary to achieve better performance. We will also add visualized results to the relevant parts of the appendix.
> > >
> > > Here we described the additional results in detail: We get the CQL, BCQ, and TD3-BC baselines in the Gym environments to mask 50% of the frames in the dataset. The results fall into one of these four categories: A. the masking operation significantly hinders the performance, where the zero frame-dropping performance goes down to nearly zero. B. masking frames makes the performance worse than the orginal baseline by a margin. C. there's no profound difference or comparable between the baseline with/without random frame dropping. But we note that all the performance is still inferior to DeFog. we summarize the distribution of these experiments' performance in the following table for better illustration:
> > >
> > > | Methods \ Performance | A. near zero | B. relatively worse | C. no difference or comparable |
> > > | ------ | ------------ | ------------------- | ---------------- |
> > > | TD3-BC | 3/9          | 1/9                 | 5/9              |
> > > | IQL    | 4/9          | 2/9                 | 3/9              |
> > > | CQL    | 3/9          | 0/9                 | 6/9              |

---

> ### Author Response · Authors · 2022-11-17
> **Response to Reviewer dgXg(3/4)**
>
>
> ### Clarity
>
> **Q**: *“ … it is not immediately clear whether the authors consider frame dropping during data collection, training, finetuning, at test time or a combination of these. I recommend adding more details about the precise setting in the introduction to avoid confusion. The experimental setup should be further clarified since it's not very clear. …”*
>
> **A**: We thank the reviewer for the recommendations to further clarify our settings. We use the widely adopted datasets such as D4RL and atari-replay-dataset which do not contain any dropped frames. During train time, we used masked frames to implement train-time frame dropping. Specifically, a random mask would effectively use all the information contained in the dataset and a fixed mask would lead to never-seen information, mimicking a corrupted dataset. The drop rate $p$ is the probability of every frame being dropped, so it is also the expected proportion of frames dropped. It is either constant or increases linearly throughout training if we use periodic resampling, as mentioned in the last paragraph of Section 3.2.2. We use a separate drop rate for finetuning, which stays constant until finetuning ends. Evaluations are online for all methods, and we add a wrapper to the environment that discards the observation by a given drop rate $p$. The results shown in Figures 3 and 4 are the average returns of the agents under different drop rates, and for every evaluated trajectory the drop rate is fixed to a specific value, as mentioned in the third paragraph of Section 4.1.
>
> ---
>
> **Q**: *“The authors mention making deep RL (DRL) agents robust to frame dropping in the abstract and introduction, but this is misleading since they don't propose a DRL algorithm, but rather an algorithm that learns from offline datasets which is a very different settings. Hence, those claims are incorrect.”*
>
> **A**: Since DeFog is based on Decision Transformer[4], we follow the terminology conventions of recent transformer-based offline RL methods such as the Decision Transformer and Trajectory Transformer[5], which are described as offline RL algorithms in the original papers.
>
> ---
>
>
> **Q**: *“On page 2, you mention the drop-span embeddings but don't define them. I don't think this is a common term in the literature so I suggest defining it the first time you mention it to avoid confusion.”*
>
> **A**: Thank you for pointing this out. We have added a definition of drop-span embeddings in Section 1.
>
> ---
>
> **Q**: *“Other details such as the number of runs used to compute the results in the plots are missing. This is an important detail so should be mentioned in the main paper rather than appendix.”*
>
> **A**: Thank you for pointing this out. We train our models on 3 different seeds, and evaluate in the environment for 10 trials. The results plotted in Figures 3, 4, and 7 are averaged on the 3 seeds. We have updated the number of seeds and runs in Section 4.1 of our paper.
>
> ---
>
> **Q**: *“I found Figure 2 to be somewhat hard to read. I'm not sure why you chose to add those lines on top of the timestep and drop-span embeddings. If they don't serve a particular purpose, I suggest removing them to make it more readable.”*
>
> **A**: Thank you for your advice, we have removed the lines in Figure 2 for better readability.
>
> ---
>
> **Q**: *“I found the sentence on page 3 "the optimization goal of our method is to imitate the oracle policy that have access to the true state in a frame-dropping environment through learning from the trajectories generated by the oracle" quite confusing. It seems to be contradicting your motivation of learning under frame-rating which would imply that you don't have access to the oracle and perfect trajectories, but rather the data was collected under frame dropping scenarios. I don't know when this assumption is realistic since it's as if you're saying "we can't collect uncorrupted data but here we assume we have access to uncorrupted data".”*
>
> **A**: As we have explained, DeFog is a general framework that can be used when the offline dataset is corrupted or not. The goal of DeFog is to extract useful information from the offline dataset so that the agent can act smoothly and safely in a frame-dropping environment. We have rewritten this sentence in our revised paper to avoid potential confusion. Thank you for the suggestion.

---

> ### Author Response · Authors · 2022-11-17
> **Response to Reviewer dgXg(2/4)**
>
> **Q**: *“... I don't think the comparison between online RL methods like TD3 / RLRD and DT / DeFog is fair. DT and DeFog have access to the oracle trajectories without any dropped frames (which DeFog perturbs to make the policy more robust). If I understand correctly, TD3 and RLRD collect data online under the frame-dropping regime so they can never see the same trajectory with and without certain frames dropped in order to learn what it should do in that situation. To make it fairer, you need to basically collect the offline dataset used to train DT and DeFog under the same frame-dropping setting/environment as used for training your online RL baselines.”*
>
> **A**: For the fairness between DeFog and other baselines, we have mentioned in Section 4.1 that the TD3 baseline is trained under regular non-frame-dropping settings, thus having the privilege of interacting with the environment without being hindered by frame dropping during training. While RLRD is trained under a frame-dropping setting, the original algorithm is designed for these settings.  It also uses an augmented state containing an action buffer and the time spans of delay for every observation frame, thus we believe RLRD is not disadvantageous compared to DeFog. For the offline baselines that we have later added, we train them with perfect unmasked datasets. We explain below the reason for doing so.
>
> ### Further Comments Beyond this Question
>
> We train the offline baselines with perfect unmasked datasets. While the reviewer suggests that using a masked, corrupted dataset would help the agent familiarize the frame dropping setting, we find that state-of-the-art methods like TD3+BC[1] still fail to keep performance under high frame drop rates even when trained with corrupted datasets, sometimes resulting in disastrous performance. We added the results to the "Training a non-Decision Transformer Model on Masked-out Datasets" part in Appendix C.1 of our paper. We believe this is due to these methods not being designed with any mechanism to process the drop-span information. We, therefore, train these methods in perfect datasets and conditions, which make them advantageous over DeFog.
>
> ---
>
> **Q**: *“In addition, DeFog has explicit access to the timestep and drop-span embedding. Do TD3 and RLRD have access to this information? There is no mention of this but if they don't, the comparison is unfair because DeFog again has privileged information.”*
>
> **A**: In principle, all the methods have access to the temporal information. The timestep and drop-span embedding are specific representation of these information as a part of DeFog. For example, the RLRD method uses an augmented state space which contains an action buffer that has the length of the maximum possible delay, as well as three different kinds of delay values, which are different forms of drop-span information. TD3 does not target at frame dropping scenarios by design so it does not process drop-span in an explicit way.
>
> ---
>
> **Q**: *“I would also like to see ablations of DeFog that don't use the drop-span embedding at all, one that doesn't use the timestep embedding at all, and one that doesn't use any of them. This would help in better assessing the contribution of each part of the architecture. Otherwise, again it is difficult to know where the gains are coming from.”*
>
> **A**: As this ablation experiment was also requested by other reviewers, we give the results and discussions in the “Drop-Span and Timestep Embedding” section of our general response. In short, drop-span embedding proved to be very important, while timestep embedding was less influential.
>
> ---
>
> **Q**: *“For the finetuning part, I would like to see ablations that only finetune the action predictor and only finetune the drop-span encoder.”*
>
> **A**: We have done this ablation and provided results in the “Finetuning Individual Components of DeFog” section of our general response. In short, finetuning only the action predictor or drop-span encoder did not outperform our current approach of finetuning both.
>
> ---
>
> **Q**: *“It would also be interesting to see what happens if instead of repeated prior frames, you replace them with zeros or an average of the N previous frames, a random frame from the previous N, or the previous frame with some added noise.”*
>
> **A**: We have conducted further ablation experiments on replacing the dropped frame with several different kinds of tokens, including zero tokens and adding noise to previous frames. We find that the performance decreases. We provide further details and results in the “Placeholder for Dropped Frames” section in our general response, as other reviewers have asked for similar experiments.

---

> ### Author Response · Authors · 2022-11-17
> **Response to Reviewer dgXg(1/4)**
>
> Dear **Reviewer dgXg**, we thank you for your detailed and thorough review. In the following sections, we seek to address each of your concerns.
>
> ---
>
> **Q**: *“… the whole point of solving this realistic setting is that you cannot collect / don't have access to such perfect / uncorrupted trajectories, isn't it? To me it seems like you motivated your work with one setting but you proceed to solve a completely different setting with totally unrealistic assumptions for the problem you claim to solve. ”*
>
> **A**: DeFog is a general framework to deal with frame dropping effects with either a dataset of uncorrupted or corrupted trajectories. We argue that both settings are practical and reasonable. Admittedly, when you are controlling a drone in a deep forest, the transmitted data can be corrupted. In this case, we might only assume access to the corrupted datasets. However, even in the same scenario, if the drone can record observations and actions and fly back with the data, the data would contain perfect trajectories. In a more common setting, an autonomous robot/drone/vehicle navigating in the city may collect perfect trajectories easily. But when there are power issues or communication issues, they need to deal with dropped frames. Empirical results show that DeFog is effective in both settings. The results and detailed description are shown in the answer to the following question.
>
> ---
>
> **Q**: *“I believe the setting you should be considering (as motivated by the intro) is one in which your offline dataset contains dropped frames but your goal is still to learn a robust policy from them, perhaps by randomly masking out other frames. In addition, you seem to be changing the dropped frames as mentioned at the top of page 5 "sample drop-mask periodically from it". Again, this indicates that your method has access to oracle trajectories collected without any frame dropping and then you randomize which ones are dropped, which of course will make the method more robust.”*
>
> **A**: We conducted the experiment with datasets in which over 60% of the frames are corrupted. During training, the corrupted frames are never seen by Defog. Results show that the performance of DeFog is similar to using the full dataset and still outperforms other baselines. The return curve against frame drop rate is added to Appendix C.1 in our revised paper.
>
> ---
>
> **Q**: *“It is not surprising at all that if you mask out observations you will learn a more robust policy than DT which learns with full observations so when observations are masked out it has no way of knowing how to deal with that.”*
>
> **A**: We agree that masking out the training data is a straightforward approach to improve the robustness of an agent against frame dropping. However, it is the integrated framework rather than the raw idea that enables the agent to succeed in challenging frame dropping scenarios. For example, as shown in Figure 7, only masking out the training data is far from sufficient to resolve this challenging task. We propose drop-span embedding and freeze trunk finetuning along with the data masking idea, both of which are also important.
>
> ---
>
> **Q**: *“In general, comparing with online RL is quite strange and hard to make a fair comparison because it collects its own data so the results will be highly dependant on the offline dataset used by DT / DeFog (even assuming that your offline data is collected under the same conditions as the online data). Given the above, the paper is incomplete without more fair comparisons with strong offline RL baselines trained on the same datasets as DT / DeFog such as IQL, CQL, offline DQN, and even BC (behavioral cloning) or BC transformer. These experiments could shed more light into whether the benefits are coming from the transformer architecture, offline learning, masking approach, dataset or something else. It could be the case that this masking also help offline RL methods which would make your approach even more general.”*
>
> **A**: Our offline datasets (D4RL and atari-replay-datasets) were all collected by online RL agents under non-frame-dropping conditions, and were divided into subsets based on the average returns achieved by the online agent. While DeFog’s performance did change according to which dataset it was trained on, its best average returns are similar to those of the online RL agents when the test-time frame drop rate is set to 0. To further analyze the source of improvements in performance, we have added offline RL baselines including CQL[2], BCQ[3], TD3+BC[1]. We find that these offline RL baselines all face a huge performance hit as the frame drop rate increases. We add the results to the “More Baselines for Continuous and Discrete Control Tasks” section of our general response and Figures 3 and 4 of our paper.

---

### Official Review · Reviewer_qcaZ · 2022-10-25

**Confidence:** 3
**Correctness:** 3
**Technical Novelty And Significance:** 2
**Empirical Novelty And Significance:** 3
**Recommendation:** 6

**Clarity, Quality, Novelty And Reproducibility:**

# Novelty

The frame dropping is easy-to-implement and effective in some cases. I can expect that this simple trick would be used widely in the community

# Reproducibility

I have mixed feelings about the reproducibility: even though the frame dropping is easy to implement, I still hope the authors could release the complete experimental code for others to reproduce.

# Requested Changes

- More baselines for discrete control task settings
- Analyzing why finetuning on the discrete control setting doesn't work
- Discussing the placeholder of dropped frames

**Strength And Weaknesses:**

# Strength

- The problem is well defined, with clear mathematical formulation, i.e., Random Dropping Markov Decision Process.
- This method is quite simple, with just changes on the input layer and training strategy from vanilla Offline Decision Transformer;
- This method is quite effective, with convincing results on well-known platforms compared to strong baselines.

# Weakness

- This submission ignored a very related paper “The Sensory Neuron as a Transformer: Permutation-Invariant Neural Networks for Reinforcement Learning, NeurIPS 2021”. Although the NeurIPS paper does not consider dropping frames and the motivation is different, the effect of random replacement of the input signals is similar to frame dropping in my opinion.
- The experiments are not enough in terms of discrete control tasks: (1) the baselines are strong enough and not representative; (2) the performance difference is not that noticeable compared to the offline Decision Transformer baseline; (3) DeFog outperforms offline Decision Transformer even if the frame drop rate is 0.00, which suggests that the random frame dropping strategy brings performance gains as well in scenarios without frame dropping problem. This lacks further discussion; (4) DeFog/f (with finetuning) is not discussed in this setting due to zero improvement. This lacks further discussion as well.
- There are too few ablation experiments that only contain the part about drop-span embedding. The supplemental discussions about (at least) drop-mask distribution beyond Bernoulli Process would be interesting.
- As for the ablation study: only drop-span embedding v.s. implicit embedding is not adequate. It is interesting to ask what would happen if we simply replace the dropped frame with a special token like [MASK] instead of with the last tokens.

**Summary Of The Paper:**

This paper targeted at an interesting and important problem: controlling the agents against frame dropping. Although there are some works aiming to solve that, they are either bottlenecked by delay threshold, or not built on the recent Decision Transformer backbone. This paper proposed a simple yet effective pre-training strategy with Offline Decision Transformer. That is, pre-training Decision Transformer with random frame dropping. This work is in line with well-known self-supervised learning methods such as Masked Language Modeling, etc. The decision robustness against frame dropping is enhanced with minimal modeling changes. The experiments on MuDojo and Atari environments (both continuous control and discrete control settings) demonstrated the effectiveness.

**Summary Of The Review:**

This submission proposes frame-dropping to improve the robustness of decision transformers. However, the core idea is too similar to a previous work “The Sensory Neuron as a Transformer: Permutation-Invariant Neural Networks for Reinforcement Learning, NeurIPS 2021”.

---

> ### Author Response · Authors · 2022-11-17
> **Response to Reviewer qcaZ(2/2)**
>
> **Q**: *“The supplemental discussions about (at least) drop-mask distribution beyond Bernoulli Process would be interesting.”*
>
> **A**: Thank you for your comment. We go beyond the Bernoulli Process, and conduct experiments on the frame dropping being a Markov process. The probability of the next frame being dropped is no longer a constant value $p$, but instead follows the matrix $P$
>
> $$
> \begin{bmatrix}
> 1 - p_1 & p_1 \\\\
> 1 - p_2 & p_2
>  \end{bmatrix}
> $$
>
> where given that the current frame was not dropped, the probability of the next frame to be dropped is $p_1$; if the current frame was dropped, then the probability of the next frame being dropped is $p_2$. The reason for choosing a Markov process is that it resembles the behavior of communication errors where frames are dropped chunk by chunk rather than frame by frame, thus simulating a typical category of realistic frame dropping issues. If $p_1 = p_2$, then the situation is identical to a Bernoulli process.
>
> Our experimental results, given in Appendix C.2, show that when comparing the Bernoulli to the Markov, the $p$ in Bernoulli is highly correlated with the $p_2$ in Markov. We find this result to abide with the fact that in a frame dropping setting, the moments where frames are dropped affect the overall performance more. If we fix $p_2$ and change $p_1$, we find that in general the less $p_1$ is, the better the performance. This is not surprising as $p_1$ decreasing would imply that there are more timesteps of consecutive undropped frames where the agent can make better decisions.
>
> ---
>
> **Q**: *“As for the ablation study: only drop-span embedding v.s. implicit embedding is not adequate. It is interesting to ask what would happen if we simply replace the dropped frame with a special token like [MASK] instead of with the last tokens.”*
>
> **A**: To further analyze the functionality of the drop-span embedding, we conduct ablation experiments on replacing the dropped frame with several different kinds of tokens, including a learnable [MASK] token. In short, we find that the performance deteriorated with such a replacement. We provide details and results in the “Placeholder for Dropped Frames” section in the general response, as other reviewers have asked for similar experiments.
>
> ---
>
> **Q**: *“I have mixed feelings about the reproducibility: even though the frame dropping is easy to implement, I still hope the authors could release the complete experimental code for others to reproduce.”*
>
> **A**: Thank you **Reviewer qcaZ**, we provide a link to our anonymous codebase and describe our plan to open source DeFog’s code in the “Code and Reproducibility” section of the general response.
>
> ---
>
> ### Requested Changes
>
> - *“More baselines for discrete control task settings”*
> - *“Analyzing why finetuning on the discrete control setting doesn't work”*
> - *“Discussing the placeholder of dropped frames”*
>
> **A**: As the other reviewers also asked for similar experiments, we provide the results, details, and analyses in the general response section. We have added more baselines for both discrete and continuous; finetuning was further ablated and analyzed; three different kinds of placeholders of dropped frames were experimented on.
>
> ---
>
> ### References
>
> [1] Y. Tang and D. Ha, “The Sensory Neuron as a Transformer: Permutation-Invariant Neural Networks for Reinforcement Learning.” arXiv, Sep. 28, 2021. [Online]. Available: http://arxiv.org/abs/2109.02869
>
> [2] A. Kumar, A. Zhou, G. Tucker, and S. Levine, “Conservative Q-Learning for Offline Reinforcement Learning.” arXiv, Aug. 19, 2020. [Online]. Available: http://arxiv.org/abs/2006.04779
>
> [3] S. Fujimoto, D. Meger, and D. Precup, “Off-policy deep reinforcement learning without exploration,” in Proceedings of the 36th international conference on machine learning, Jun. 2019, vol. 97, pp. 2052–2062. [Online]. Available: https://proceedings.mlr.press/v97/fujimoto19a.html
>
> [4] S. Fujimoto and S. S. Gu, “A Minimalist Approach to Offline Reinforcement Learning.” arXiv, Dec. 03, 2021. Accessed: Nov. 14, 2022. [Online]. Available: http://arxiv.org/abs/2106.06860

---

> > ### Comment · Reviewer_qcaZ · 2022-11-25
> > **Increasing score to 6**
> >
> > Thanks for the responses.
> >
> > I've read the authors' responses and other reviewers' comments. I appreciate the authors' efforts to continuously improve the quality of this submission based on our suggestions and discussions. Therefore, I decide to increase the score from 5 to 6.

---

> > > ### Author Response · Authors · 2022-11-28
> > > **Further Discussion**
> > >
> > > Dear reviewer qcaZ, thank you for your discussion and suggestions that make our work more comprehensive, and we will continue to polish it for the final version.

---

> ### Author Response · Authors · 2022-11-17
> **Response to Reviewer qcaZ(1/2)**
>
> Dear **Reviewer qcaZ**, we thank you for your detailed and thorough review. In the following sections, we seek to address each of your concerns.
>
> ---
>
> **Q**: *“This submission ignored a very related paper “The Sensory Neuron as a Transformer: Permutation-Invariant Neural Networks for Reinforcement Learning, NeurIPS 2021”. Although the NeurIPS paper does not consider dropping frames and the motivation is different, the effect of random replacement of the input signals is similar to frame dropping in my opinion.”*
>
> **A**: In “The Sensory Neuron as a Transformer: Permutation-Invariant Neural Networks for Reinforcement Learning”[1], the authors use separate but similar neural network models to process local information and use an attention mechanism to integrate them together and generate a global latent code for the agent’s decision making. In our humble opinion, this paper is only similar to ours on a higher level because we all perform some sort of corruption or permutation on the data to level up the problem, thus improving the robustness of the agent. Meanwhile, the actual approach we use are different: 1) we aim towards a totally different problem setting of frame dropping; 2) we still abide to the Decision Transformer’s method of feeding trajectories by a sequential matter, not permuting the order; 3) we explicitly encode the frame dropping information into the model by drop-span embedding. Nonetheless as this was a previous paper we didn’t step upon, we have added it into our Related Works section.
>
> ---
>
> **Q**: *“[for Atari] the baselines are not strong enough and not representative;”*
>
> **A**: We add offline RL baselines for the MuJoCo and the Atari environments, including CQL, BCQ, and TD3+BC. DeFog is able to outperform these baselines under frame dropping conditions. We provide further details in the “More Baselines for Continuous and Discrete Control Tasks” section of the general response.
>
> ---
>
> **Q**: *“[for Atari] the performance difference is not that noticeable compared to the offline Decision Transformer baseline;”*
>
> **A**: We believe this is due to the various differences between the MuJoCo and Atari  environments, including the transition dynamics, the high dimensionality observation space, as well as some other reasons. As the other reviewers have asked similar questions, we provide a detailed analysis and explanation in the "Discussion on the Atari Environments" section of the general response.
>
> Despite these problems, DeFog is still able to achieve better performance than a number of offline baselines in frame dropping scenarios. In the updated version of our paper, two more common offline RL methods, CQL and BCQ are proven to be weaker than DeFog as well. It’s worth noting that our method also provide a framework for solving frame dropping problem, and could be applied onto other models beyond the Decision Transformer to overcome these challenges.
>
> ---
>
> **Q**: *“[for Atari] DeFog outperforms offline Decision Transformer even if the frame drop rate is 0.00, which suggests that the random frame dropping strategy brings performance gains as well in scenarios without frame dropping problem. This lacks further discussion;”*
>
> **A**: We believe that in some of the environments and tasks, using masked out datasets not only helps the agent to be more robust to frame dropping, but also makes the task more challenging in the sense that the agent needs to understand the environment dynamics to give better action predictions. This requirement not only helps the agent in frame dropping scenarios, but also lets it make better decisions in general, help to improve the  performance when frame drop rate is 0. We add this discussion to Section 4.3 of our paper.
>
> ---
>
> **Q**: *“[for Atari] DeFog/f (with finetuning) is not discussed in this setting due to zero improvement. This lacks further discussion as well.”*
>
> **A**: In practice, the freeze trunk method is still applied to the model in the Atari environment, and the final results are selected from the finetuned models. However, due to the relatively high variance in Atari environments, the effectiveness of the finetuning scheme is not that ubiquitous across MuJoCo environments, thus we omit this part of discussion.

---

### Official Review · Reviewer_4AHJ · 2022-10-27

**Confidence:** 4
**Correctness:** 3
**Technical Novelty And Significance:** 2
**Empirical Novelty And Significance:** 3
**Recommendation:** 6

**Clarity, Quality, Novelty And Reproducibility:**

This work is well motivated and the writing is easy to follow. I like it starts with the real-world challenges and then formulate the research problem based on it. The proposed approach is simple yet effective. Empirical results validate the robustness of Defog against frame-dropping issues in most of continuous control tasks when comparing with the other approaches.
Despite only empirical comparisons, I like this paper also trying to show some insights underlying the technical approach, e.g. visualization analysis.

However, the technical novelty is limited. The propose approach seems a simple extension on DT, specifically adding random mask to simulate the frame-dropping and a drop-span embedding to give explicit signals to the model. While the whole model architecture and the training objectives are all the same with DT. Empirically, Defog demonstrated its effectiveness on MuJoCo. While on Atari, seems it does not show much benefits when comparing with a vanilla DT.

Besides, no source codes are provided, so I can not validate its reproducibility.

**Strength And Weaknesses:**

Pros:
This work is well motivated and the writing is easy to follow. I like it starts with the real-world challenges and then formulate the research problem based on it.
The proposed approach is simple yet effective. Empirical results validate the robustness of Defog against frame-dropping issues in most of continuous control tasks when comparing with the other approaches.
Despite only empirical comparisons, I like this paper also trying to show some insights underlying the technical approach, e.g. visualization analysis.

Cons:
My main concern is the technical novelty. The propose approach seems a simple extension on DT, specifically adding random mask to simulate the frame-dropping and a drop-span embedding to give explicit signals to the model. While the whole model architecture and the training objectives are all the same with DT.
Another concern is that the drop-span embedding has a strong dependence on knowing how where and how many frames are dropped. However, in many real-world applications, such signals are unknown, in these cases the drop-span beddings limited the broad applications of Defog.
Empirically, Defog demonstrated its effectiveness on MuJoCo. While on Atari, seems it does not show much benefits when comparing with a vanilla DT.

Minor:
Typo Sec. 4.4 'DeFogin' -> 'DeFog in ....'

Some questions:
1) Would you give more analysis on the different observations on continuous tasks (Mojoco) vs, discrete tasks (Atari)?
2) it is good see the ablation on the contribution of drop-span embedding and finetuning stage. While that would be good to give more insights on why and how such training regime can improve the robustness.
3) I am curious what if Defog also randomly mask out actions. Because the current training gives the model ground actions sequentially, which maybe lead to trivial solutions as the model already seen the actions signals for next step prediction. In Fig 5 and Fig 6, we can see that the trained defog tends to give smooth actions which maybe adapted to the showing cases. But if an agent has drastic motion pattern, then naively insert smooth actions would collapse the agent.
4) what if also ask the model to reconstruct the masked out states?

**Summary Of The Paper:**

This paper presents Decision Transformer under Random Frame Dropping (DeFog). Defog is an simple yet effective algorithm based on DT that is designed to robust to frame dropping issue, which is one of the key challenges in real-world remote control applications. Defog simulates the frame dropping phenomenon  by randomly masking out the observations in offline datasets and embeds frame dropping timespan information explicitly to the model. Experiments are conducted on both continuous and discrete control tasks, i.e. MuJoCo and Atari. Empirical results show that Defog is robust to frame-dropping issue in some of the tasks, mainly on MuJoCo.

**Summary Of The Review:**

This work is well motivated and the writing is easy to follow. I like it starts with the real-world challenges and then formulate the research problem based on it. The proposed approach is simple yet effective. Empirical results validate the robustness of Defog against frame-dropping issues in most of continuous control tasks when comparing with the other approaches.
Despite only empirical comparisons, I like this paper also trying to show some insights underlying the technical approach, e.g. visualization analysis.

However, the technical novelty is limited. The propose approach seems a simple extension on DT, specifically adding random mask to simulate the frame-dropping and a drop-span embedding to give explicit signals to the model. While the whole model architecture and the training objectives are all the same with DT. Empirically, Defog demonstrated its effectiveness on MuJoCo. While on Atari, seems it does not show much benefits when comparing with a vanilla DT.  Besides, no source codes are provided, so I can not validate its reproducibility.

---

> ### Author Response · Authors · 2022-11-17
> **Response to Reviewer 4AHJ(2/2)**
>
> **Q**: *“I am curious what if Defog also randomly mask out actions. Because the current training gives the model ground actions sequentially, which maybe lead to trivial solutions as the model already seen the actions signals for next step prediction. In Fig 5 and Fig 6, we can see that the trained defog tends to give smooth actions which maybe adapted to the showing cases. But if an agent has drastic motion pattern, then naively insert smooth actions would collapse the agent.”*
>
> **A**: We would first like to clarify that we use causal masks on states, actions, and reward-to-goes so the model does not have any access to future actions. As the action taken is determined by the DeFog agent itself, the past action history is always visible to the model, so we believe that past actions do not need to be masked out in training for practical applications. Nonetheless we perform an experiment where actions are masked out alongside the states and reward-to-goes, and results show that the performance is negatively affected. We add this experiment to our Appendix C.2.
>
> ---
>
> **Q**: *“what if also ask the model to reconstruct the masked out states?”*
>
> **A**: Thank you for your advice. Although the authors of the Decision Transformer have stated that adding reconstruction loss over the states and reward-to-goes does not improve its performance emperically, reconstructing the states could still benefit the model in a frame dropping setting since the real state is usually hidden. To verify this hypothesis, we conduct experiments on nine MuJoCo settings to reconstruct the state and the rewards-to-go separately and together. Our empirical results show that reconstructing those tokens together have little or no impact on DeFog’s performance, whether in a frame dropping setting or not. However, if the model is asked to reconstruct the masked out states only, its performance deteriorates on several high frame-dropping settings. One potential explanation is that the unbalanced loss setting over the states and the reward-to-goes make the model less aware of the reward signal, thus hindering it’s ability for decision making in challenging environments. We provide further details of the experiment in the “Reconstruction of Frames During Training” part of our Appendix C.1.
>
> ---
>
> **Q**: *“Besides, no source codes are provided, so I can not validate its reproducibility.”*
>
> **A**: Thank you **Reviewer 4AHJ**, we provide a link to our anonymous codebase and describe our plan to open source DeFog’s code in the “Code and Reproducibility” section of the general response.
>
> ---
>
> ### References
>
> [1] V. Tangkaratt, N. Charoenphakdee, and M. Sugiyama, “Robust Imitation Learning from Noisy Demonstrations.” arXiv, Feb. 19, 2021. [Online]. Available: http://arxiv.org/abs/2010.10181

---

> ### Author Response · Authors · 2022-11-17
> **Response to Reviewer 4AHJ(1/2)**
>
> Dear **Reviewer 4AHJ**, we thank you for your detailed and thorough review. For similar questions coming from different reviewers, we have replied to them in the general response. In the following sections, we seek to address each of your concerns.
>
> ---
>
> **Q**: *“My main concern is the technical novelty. The propose approach seems a simple extension on DT, specifically adding random mask to simulate the frame-dropping and a drop-span embedding to give explicit signals to the model. While the whole model architecture and the training objectives are all the same with DT.”*
>
> **A**: While DeFog is based on the Decision Transformer, we focus on a practical setting of frame dropping that previous works have under-explored. Meanwhile, the train-time frame-dropping and drop-span embeddings are methods that could be easily extended onto different model structures. Thus, we provide pathways for solving the frame dropping issue in a general sense, without largely extending the time needed for training. While DeFog agents are able to perform much better compared to baselines under severe frame dropping conditions, the performance in non-frame-dropping scenarios is also kept or improved.
>
> ---
>
> **Q**: *“Another concern is that the drop-span embedding has a strong dependence on knowing how where and how many frames are dropped. However, in many real-world applications, such signals are unknown, in these cases the drop-span beddings limited the broad applications of Defog.”*
>
> **A**: We agree with the reviewer that the number of frames dropped is an important piece of information for DeFog to perform well in frame-dropping environments. Meanwhile, signals containing such information can be delivered through several methods. For example, a common practice in network protocols is to send a timestamp alongside the observation frame’s data pack. So once received, the drop-span could be calculated. If the sensor itself malfunctions and doesn’t provide any observation, the agent would also know a frame was dropped. In addition, there are many works in detecting noisy observation frames in control environments, for example [1].
>
>
> ---
>
> **Q**: *“Empirically, Defog demonstrated its effectiveness on MuJoCo. While on Atari, seems it does not show much benefits when comparing with a vanilla DT. Would you give more analysis on the different observations on continuous tasks (Mojoco) vs, discrete tasks (Atari)?”*
>
> **A**: We believe that the differences between DeFog’s performance on MuJoCo vs. Atari are due to many reasons, including the high dimensional observation space, the different transition dynamics, and other reasons. We provide a detailed explanation and analysis in the "Discussion on the Atari Environments" section of the general response, as this is also the interest of other reviewers.
>
> Despite these problems, DeFog is still able to achieve better performance than a number of offline baselines in frame dropping scenarios. In the updated version of our paper, two more common offline RL methods, CQL and BCQ are proven to be weaker than DeFog as well. It’s worth noting that our method also provide a framework for solving frame dropping problem, and could be applied onto other models beyond the Decision Transformer to overcome these challenges.
>
> ---
>
> **Q**: *“it is good see the ablation on the contribution of drop-span embedding and finetuning stage. While that would be good to give more insights on why and how such training regime can improve the robustness.”*
>
> **A**: We have performed more experiments on drop-span embedding and finetuning. In short, we find that the information a DeFog agent really needs is the number of past timesteps before a frame was observed, not the timestep it was observed, thus it is important to explicitly encode this number into the model.  For finetuning, the action predictor and the drop-span encoder are both key components of the DeFog model, one giving the drop-span information, and the other deciding what action to take. We provide further details and analyses in the “Drop-Span and Timestep Embedding” and “Finetuning Individual Components of DeFog” sections of the general response.

---

> ### Author Response · Authors · 2022-12-07
> **Further Discussion**
>
> Dear reviewer 4AHJ,
> We would like to first thank you again for your constructive comments and helpful suggestions. Since we are nearly at the end of the discussion phase, we would like to post a follow-up discussion.
> In our previous response, we have clarified the raised questions and made corresponding improvements in the updated paper.
> We want to point out that DeFog is the first paper to address the frame-dropping problem in the field of deep reinforcement learning. While our modification to the Decision Transformer model is straightforward, the performance gain is significant. We believe this kind of simplicity could encourage others to pay more attention to this vital domain. Also, the proposal of drop-span embedding demonstrates the effectiveness of providing additional time-related information alongside the positional encoding in transformers, which could also benefit others in a variety of situations.
> We hope to discuss further with you whether your concerns have been addressed or not. If you still have any unclear parts of our work, please let us know.

---

### Author Response · Authors · 2022-11-17
**General Response(1/2)**

Dear reviewers, we appreciate all your helpful feedback. In this response, we address the comments and questions that are in common. We welcome further discussion with each of the reviewers to address any remaining concerns.

We’d like to thank **Reviewer 4AHJ** for acknowledging that *“DeFog is a simple yet effective algorithm”* that solves “one of the key challenges in real-world remote control applications”, and **Reviewers qcaZ, dgXg** think that our method *“would be used widely in the community”* and is tackling a *“well-motivated problem of some practical interests”*. We believe that with the help of your constructive advice, DeFog is able to become a more solid and useful algorithm for the community.

We want to reiterate that DeFog aims to solve one important yet largely unexplored problem in real-world scenarios, that is, to control against random frame dropping of observations. Our method could be seen as an out-of-the-box solution to the drop-frame problems with moderate modification to an offline Decision Transformer model, and could significantly improve performance on a wide range of continuous and discrete control tasks in the drop-frame setting while keeping the non-drop-frame performance unaffected.

### Placeholder for Dropped Frames

We thank the reviewers for pointing out that the choice of what to use in replacement for dropped frames may be an influential factor to DeFog’s performance. 	We conduct further ablation studies by replacing the dropped frame with the following settings:

- As per **Reviewer dgXg**’s suggestion, adding noise to the dropped frames. This could be interpreted as stimulating the evolution of the unknown real states. For each noise step, we sample from a Gaussian distribution that is drawn from all the changes between consecutive observations from the dataset. When multiple frames are dropped, those Gaussian noises are added to form a new Gaussian distribution. However, despite this perturbation in observation, the overall performance of our methods has no significant change. We used a noise coefficient of both 0.1 and 0.5 to experiment on the scale of the noise, and neither settings helped Defog’s performance.We find that increasing the noise simply deteriorates performance, possibly due to the noisy observation distracting the agent. In datasets such as Hopper-medium and Walker-expert, the deterioration was more noticeable.
- We follow the advice of **Reviewer qcaZ** to replace the embedding of those replaced tokens to a specific learnable [MASK] token. We tried 2 settings, one where the dropped observation and dropped reward-to-go shares the same [MASK] token, and one where the two tokens are separate. Empirical results show that both settings give performances better than vanilla Decision Transformer but worse than DeFog. We do not find this surprising as a single learnable mask cannot carry enough information for all the dropped frames, while the previous frame that DeFog uses would usually be similar to the current dropped frame.
- We follow another suggestion of **Reviewer dgXg** to replace the dropped frames with zeros. The results show that performance is not that much better than the vanilla Decision Transformer. As the zero token basically provides no information, this aligns with our expectation that it should perform worse than using the learnable [MASK] token as replacement for dropped frames.

Detailed results of these experiments can be found in the "Placeholder for Dropped Frames" part of Appendix C.2 in our paper.


### Drop-Span and Timestep Embedding

To further illustrate the importance of DeFog’s drop-span embedding, we perform experiments on removing it. We have added the results of removing drop-span embedding from DeFog to the Appendix C.3. Results show that performance is similar to adding the implicit drop-span encoder. We believe this shows that the actual useful information leveraged by a DeFog agent is **the number of past timesteps before** a frame was observed, not **the timestep** it was observed. The longer the drop-span of the frame, the less it should be considered in the action prediction process, and the action history could be a better reference for decision-making. From the similarity of implicit vs. no embedding scores we also find that useful numbers like the drop-span need to be explicitly given, and performance would be hindered even if the agent can work out the number by simple arithmetic. We also perform the experiment of removing the timestep embedding. Results, given in Appendix C.3 as well, show that the removal of the timestep embedding does not cause a severe drop in performance.

---

> ### Author Response · Authors · 2022-11-17
> **General Response(2/2)**
>
> ### Finetuning Individual Components of DeFog
>
> DeFog currently finetunes the action predictor and drop-span encoder. For a better understanding of the finetuning stage and the specific parts, we conduct experiments on these components individually. The results, given in Appendix C.3, show that finetuning only the drop-span encoder gives a slightly better performance on Walker2d-medium-replay, Walker2d-expert, and HalfCheetah-medium-replay datasets. Meanwhile, on the other datasets, for example the Hopper medium-replay, finetuning both the drop-span encoder and the action predictor led to better results. In general neither way of finetuning significantly outperforms its counterparts. We find this understandable as the action predictor and the drop-span encoder are both key components of the DeFog model. The ablation results on the drop-span encoder show that how well the drop-span information can be leveraged by the model heavily affects the performance. It is also shown that the quality of the action predictor also directly determines the ability of the agent.
>
>
> ### More Baselines for Continuous and Discrete Control Tasks
>
> To further analyze the source of improvements in DeFog’s performance, we reproduce various strong and representative baselines for both the continuous and discrete scenarios. We have added offline RL methods including CQL[1], BCQ[2], and TD3+BC[3]. We find that these offline RL baselines all face a huge performance drop as the frame drop rate increases. DeFog outperforms them under frame dropping conditions. We have added the results of these baselines into Figures 3 and 4 of our paper.
>
> ### Discussion on the Atari Environments
>
> We believe that difference between DeFog’s performance on MuJoCo vs. Atari are due to many reasons. Firstly, the high dimensionality of the Atari games’ observation space makes recovering the dropped information substantially more difficult. In the meantime, Atari games often require the agent to perform credit assignment over long timespans, which is challenging for the Decision Transformer (only 2.5% of the human average score in the Seaquest game). It’s harder to convey additional information of dropped frames under such constraints. *
>
> Also, the transitions in the Atari environments are determined by some hidden program. That being said, the Atari agents violating the physics law (or any learnable patterns) may produce actions and observations that are hard to recover once dropped. Another issue is that a common practice for agents is to stack observation frames in Atari environments, and empirical results shows that the Vanilla Decision Transformer’s performance drops without this operation. This procedure of overlapping frames makes the important drop-span information more vague, hindering the performance under high drop rates.*
>
> Additionally, in Atari games the starting position have great influence over the performance. Since this is decided randomly over each seed, it could introduce high variance to the performance. Also, the nonphysical identity of the Atari environments could contribute to this phenomenon. We note that in all the added offline RL baselines, i.e., CQL[1] and BCQ[2], the variance of the Atari agents are high as well.
>
> Despite these problems, DeFog is still able to achieve better performance than these offline RL baselines in frame dropping scenarios. In the updated version of our paper, besides the vanilla Decision Transformer, CQL and BCQ are proven to be weaker than DeFog as well. It’s worth noting that our method also provides a framework for solving the frame dropping problem, and could be applied onto other models beyond the Decision Transformer to overcome these challenges.
>
> ### Code and Reproducibility
>
> To reproduce this paper, the link for our anonymous codebase is: https://anonymous.4open.science/r/DeFog-E55F . We will open source our code upon acceptance.
>
> ### References
>
> [1] A. Kumar, A. Zhou, G. Tucker, and S. Levine, “Conservative Q-Learning for Offline Reinforcement Learning.” arXiv, Aug. 19, 2020. [Online]. Available: http://arxiv.org/abs/2006.04779
>
> [2] S. Fujimoto, D. Meger, and D. Precup, “Off-policy deep reinforcement learning without exploration,” in Proceedings of the 36th international conference on machine learning, Jun. 2019, vol. 97, pp. 2052–2062. [Online]. Available: https://proceedings.mlr.press/v97/fujimoto19a.html
>
> [3] S. Fujimoto and S. S. Gu, “A Minimalist Approach to Offline Reinforcement Learning.” arXiv, Dec. 03, 2021. Accessed: Nov. 14, 2022. [Online]. Available: http://arxiv.org/abs/2106.06860

---

### Decision · Program_Chairs · 2023-01-20

**Decision:**

Accept: poster

**Justification For Why Not Higher Score:**

No strong champion for the paper, the reviews seem a very lukewarm in the evaluation of technical novelty.

**Justification For Why Not Lower Score:**

Robustness of decision transformer-based policies under observation corruption is an important practical problem. As such, the simple training techniques benchmarked in this paper maybe of interest to RL/Robotics communities.

**Metareview: Summary, Strengths And Weaknesses:**

Summary: A very simple extension of Decision Transformer is proposed to provide robustify the model against frame dropping, which is well motivated in real-world remote control applications. The key idea is to simply to randomly mask out the observations in offline datasets and embed frame dropping timespan information explicitly to the model. MuJoCo and Atari results show the benefits relative to a comprehensive set of baselines.

Strengths: Simplicity and effectiveness of the proposal on the benchmarks tested.

Weaknesses: Technical novelty seems quite low. The paper would be more convincing if other baselines are also reported with and without frame dropping during training for a more complete empirical evaluation.



**Note From Pc:**

if the above contains the word "oral" or "spotlight" please see: "oral" presentation means -> notable-top-5% and "spotlight" means -> notable-top-25%. As stated in our emails, we are disassociating presentation type from AC recommendations